# A Systematic Review on COVID-19 Vaccine Strategies, Their Effectiveness, and Issues

**DOI:** 10.3390/vaccines9121387

**Published:** 2021-11-24

**Authors:** Shahad Saif Khandker, Brian Godman, Md. Irfan Jawad, Bushra Ayat Meghla, Taslima Akter Tisha, Mohib Ullah Khondoker, Md. Ahsanul Haq, Jaykaran Charan, Ali Azam Talukder, Nafisa Azmuda, Shahana Sharmin, Mohd. Raeed Jamiruddin, Mainul Haque, Nihad Adnan

**Affiliations:** 1Gonoshasthaya-RNA Molecular Diagnostic & Research Center, Dhanmondi, Dhaka 1205, Bangladesh; shahad@rnabiotech.com.bd (S.S.K.); mohib@gonoshasthayakendra.org (M.U.K.); ahsan@rnabiotech.com.bd (M.A.H.); mohd.raeed@bracu.ac.bd (M.R.J.); 2Strathclyde Institute of Pharmacy and Biomedical Sciences, University of Strathclyde, Glasgow G1 1XQ, UK; Brian.Godman@strath.ac.uk; 3Division of Public Health Pharmacy and Management, School of Pharmacy, Sefako Makgatho Health Sciences University, Pretoria 0204, South Africa; 4Centre of Medical and Bio-Allied Health Sciences Research, Ajman University, Ajman P.O. Box 346, United Arab Emirates; 5Department of Microbiology, Jahangirnagar University, Savar 1342, Bangladesh; rishadirfan97@gmail.com (M.I.J.); meghla.ju@gmail.com (B.A.M.); taslima.tisha.bd@gmail.com (T.A.T.); aat@juniv.edu (A.A.T.); azmuda@juniv.edu (N.A.); 6Department of Community Medicine, Gonoshasthaya Samaj Vittik Medical College, Savar 1344, Bangladesh; 7Department of Pharmacology, All India Institute of Medical Sciences, Jodhpur 342005, India; charanj@aiimsjodhpur.edu.in; 8Department of Pharmacy, BRAC University, Dhaka 1212, Bangladesh; sharmin@bracu.ac.bd; 9The Unit of Pharmacology, Faculty of Medicine and Defence Health, Universiti Pertahanan Nasional Malaysia (National Defence University of Malaysia), Kem Perdana Sugai Besi, Kuala Lumpur 57000, Malaysia

**Keywords:** clinical trials, COVID-19 vaccines, systematic review, inactivated vaccines, mRNA vaccines, nanoparticle-based vaccines, recombinant vaccines, prime-booster strategy

## Abstract

COVID-19 vaccines are indispensable, with the number of cases and mortality still rising, and currently no medicines are routinely available for reducing morbidity and mortality, apart from dexamethasone, although others are being trialed and launched. To date, only a limited number of vaccines have been given emergency use authorization by the US Food and Drug Administration and the European Medicines Agency. There is a need to systematically review the existing vaccine candidates and investigate their safety, efficacy, immunogenicity, unwanted events, and limitations. The review was undertaken by searching online databases, i.e., Google Scholar, PubMed, and ScienceDirect, with finally 59 studies selected. Our findings showed several types of vaccine candidates with different strategies against SARS-CoV-2, including inactivated, mRNA-based, recombinant, and nanoparticle-based vaccines, are being developed and launched. We have compared these vaccines in terms of their efficacy, side effects, and seroconversion based on data reported in the literature. We found mRNA vaccines appeared to have better efficacy, and inactivated ones had fewer side effects and similar seroconversion in all types of vaccines. Overall, global variant surveillance and systematic tweaking of vaccines, coupled with the evaluation and administering vaccines with the same or different technology in successive doses along with homologous and heterologous prime-booster strategy, have become essential to impede the pandemic. Their effectiveness appreciably outweighs any concerns with any adverse events.

## 1. Introduction

The current pandemic of coronavirus disease-19 (COVID-19) caused by Severe Acute Respiratory Syndrome Coronavirus-2 (SARS-CoV-2) is the seventh human coronavirus discovered [1]. Previously, in 2002–2003, SARS-CoV and 2015 Middle-East Respiratory Syndrome Coronavirus (MERS-CoV) created outbreaks in southern China and the Arabian Peninsula. Nevertheless, both episodes were centered primarily on their origin country and did not emerge into a worldwide pandemic [2,3].

Coronaviruses are categorized into four genera: *Alphacoronavirus*, *Betacoronavirus*, *Gammacoronavirus*, and *Deltacoronavirus* [4]. *Alphacoronavirus* and *Betacoronavirus* are found in mammals, while *Gammacoronavirus* and *Deltacoronavirus* infect birds [5]. HCoV-229E and HCoV-OC43, the two human coronaviruses (HCoVs), were first identified in the 1960s [6,7]. Ten complete genome sequences were available before 2003 [8,9,10]. After the SARS-CoV epidemic, sixteen more complete genome sequences were added, including two HCoVs (NL63 and HKU1), ten mammalians, and four avian coronaviruses [11,12,13,14,15,16,17,18,19,20,21,22,23,24,25,26]. The recent pandemic caused by SARS-CoV-2 started in December 2019 in Wuhan, China [27,28,29,30]. SARS-CoV-2 belongs to the *Betacoronavirus* genera, and to the order *Nidovirales* and *Orthocoronavirinae* subfamily [31,32]. Although HCoV OC43, HKU1, 229E, and NL63 mostly create mild respiratory illness, including the common cold in patients, SARS-CoV-2 can be deadly with a moderate to high severity rate similar to SARS-CoV and MERS-CoV [1,33]. The severity of the infection and its transmission is reflected by more than 246 million people having been infected and 49 million lives lost by October 29, 2021 [34].

Usually, respiratory infections occur by transmitting virus-containing droplets or aerosols from infected individuals while talking, breathing, coughing, and sneezing. COVID-19, primarily a respiratory infection, also transmits through airborne aerosols produced from infected persons, including asymptomatic patients, although it is not confirmed if transmission occurs through airborne droplets or aerosol [35,36,37,38,39,40,41].

Upon entry into the host by binding the spike protein (S) to human angiotensin-converting enzyme 2 (hACE2), SARS-CoV-2 replicates rapidly in human lung tissue [42]. However, other than the respiratory tract cells, ACE2 receptors are present in the brain, gut, endothelium and vascular smooth muscle cells, and peripheral organs, including the kidneys and liver [43]. In view of this, along with viral pneumonia, SARS-CoV-2 can cause coagulation disorders, cardiovascular impairment, neurological manifestations including systemic and local thrombotic events, ischemic or hemorrhagic stroke, meningoencephalitis, and can cause damage to the kidney and liver [44]. Pro-inflammatory cytokines/chemokines in asymptomatic patients are lower than in symptomatic patients. As a result, the virus rapidly spreads in the pharynx and shedding before symptoms occur [38,45,46]. Because of these characteristics, the viral load of SARS-CoV-2 is significantly higher than other respiratory viruses [47,48]. In severe COVID-19 cases, hyperactivation of T-cells, especially CD8+ T-Cells, leads to the release of a higher level of interferon (IFN)-γ, interleukin (IL)-2, and tumor necrosis factor (TNF)-α. However, neutrophilia over lymphopenia finally leads to cytokine storms [49,50].

While many candidates have been proposed to prevent and treat patients with COVID-19, including hydroxychloroquine, lopinavir-ritonavir, molnupiravir, and remdesivir, to date, only dexamethasone has shown a reduction in mortality in hospitalized patients receiving respiratory support; however, there is increasing evidence for medicines such as tocilizumab as well as casirivimab and imdevimab [51,52,53,54,55,56]. Whilst vaccines are being developed and administered, the recommended approach to reduce morbidity and mortality due to COVID-19 is the instigation of lockdown and social distancing measures [57,58,59,60]. However, lockdown measures have unintended consequences. Transport restrictions, closure of clinics, and concerns among patients attending hospital clinics have resulted in increases in non-communicable diseases as well as increased morbidity and mortality among unvaccinated children [61,62,63,64,65]. Lockdown measures also have economic consequences, especially among developing countries [66]. Consequently, there is increased urgency for an effective vaccine to combat COVID-19.

Vaccine development against infectious diseases has a four-century history; however, researchers face challenges developing effective vaccines against emerging infectious diseases [67,68]. Several vaccine techniques have already been invented, including live attenuated, inactivated, recombinant technologies, adenoviral vector-based, DNA, peptide, and mRNA [69,70,71]. Recently, nanotechnology has shown new potential in vaccine development [72]. However, nanoparticle-based peptide delivery has many challenges. These include safe delivery vehicles, vaccine adjuvants, antigen stability, targeted delivery, long-time controlled release, and evasion of the immune responses [73,74,75,76,77,78,79,80]. Nevertheless, nanotechnology-based vaccines are easy to design and can be produced on a large scale compared with conventional vaccines. In this ongoing pandemic, nanotechnology and nanomedicine are seen as new therapeutic approaches that could have an appreciable clinical impact [81,82,83,84,85,86,87,88].

To develop vaccines against viruses, knowledge about their genomic and immunogenicity perspectives is essential [28]. Encouragingly, nowadays, information about the genome sequence, including codon and codon pair, junction dinucleotide and individual dinucleotide, RNA structure with several frameshift regions, and transcriptome architecture, has become more accessible for developing vaccines [2,89,90,91]. Researchers have worked hard to construct effective vaccines against SARS-CoV-2, given the urgency and the lack of effective medicines. Several vaccine candidates, including inactivated, recombinant, mRNA, and nanoparticle-based strategies, have demonstrated their potency [92,93,94,95]. Consequently, we sought to review the current vaccine candidates, focusing on their development strategy, adverse events, and effectiveness based on the published findings, as we have been aware there has been considerable misinformation regarding COVID-19 and its prevention and treatment, increasing morbidity, mortality, and costs [65,96,97,98,99,100].

## 2. Methodology

A comprehensive search was undertaken using a systematic approach following the Preferred Reporting Items for Systematic Review and Meta-Analysis (PRISMA) guideline from several online databases (i.e., PubMed, ScienceDirect, and Google Scholar) to determine original research articles on COVID-19 vaccines either in pre-clinical/non-clinical/trial phases or beyond the trial phase [101]. Studies other than full-length research articles, i.e., case reports, review articles, correspondences, and letters, were excluded. Original articles not relevant to the SARS-CoV-2 vaccine were considered ineligible and also excluded. We employed Boolean logical operators (‘AND’ and ‘OR’) using ‘Advanced’ search in PubMed and ScienceDirect selecting ‘title’/’abstract’ as well as title, abstract, or author-specified ‘keywords’, respectively, and in Google Scholar selecting ‘all entitled with appropriate keywords’, i.e., ‘COVID-19′, ‘SARS-CoV-2′, ‘novel coronavirus’, ‘n-CoV’, ‘vaccine’, ‘clinical’, ‘clinical study’, ‘clinical trial’, ‘randomized control trial’, ‘RCT’, ‘efficacy’, ‘seroconversion’, and ‘side effects’ (Appendix A). A total of eight authors (S.S.K., M.I.J., B.A.M., T.A.T., M.A.H., M.R.J., M.H., and N.A.) independently screened and assessed the articles to avoid study bias. The search was conducted until 1 September 2021, with no year and language restrictions. Fifty-nine studies (i.e., clinical trials (*n* = 29), and pre-/non-clinical trial or other related original studies (*n* = 30) were included. EndNote X9 software was used to manage the references. The result of the final study selection is based on the inclusion criteria as described in Figure 1. The quality assessment of the clinical trial studies was performed using the standard of the Quality Assessment of Controlled Intervention Studies, NIH, NHLBI [102] (Table 1).

## 3. Current Vaccine Candidates

Recent advances in bioprocess technology, genomics, structural biology, and immunopathology have significantly contributed to the speed of COVID-19 vaccine development. Researchers have used accumulative knowledge from previous vaccine candidates, and within twenty-two months of the emergence of COVID-19, developed 155 preclinical vaccine candidates with 23 emergency use authorized ones. In this review, we will discuss four different types of COVID-19 vaccine strategies (Figure 2), including
(1)inactivated,(2)mRNA,(3)viral vector, and(4)nanoparticle-based peptide vaccines.

### 3.1. Inactivated Vaccine

Inactivated vaccines are formulated by inactivating virulent particles of viruses by treating the virus particle with chemicals, including formaldehyde, β-propiolactone, ethylenimine, phenol, ascorbic acid, β-aminophenylketone, and diethylpyrocarbonate [103]. Among these inactivating chemicals, formaldehyde is currently not used to reduce the risk of incomplete inactivation [104]. Since the virus particles are inactivated, they cannot multiply after entering the human body. Consequently, they are safe for administration but need to be introduced in large amounts compared to live attenuated vaccine [104]. Besides, the particles stay in place for the immune system to recognize and process. They induce inefficient cellular and humoral immune responses, showing minimal or no long-term memory response. However, many killed virus particles and adjuvants such as aluminum hydroxide are added to the vaccine formulation to improve their efficacy [105].

The previous SARS-CoV outbreak happened from the autumn of 2002 to the spring of 2003; 8098 became infected, and 774 died in 26 countries [106,107]. There was no approved antiviral medicine against SARS-CoV to prevent such an outbreak. Two types of vaccines, the whole killed virus (WKV) vaccine and the Ad-vectored vaccine, were tested for their potency in producing neutralizing antibodies in the ferret [108]. The study results showed that the WKV vaccine induced a 15-fold higher production of neutralizing antibodies than the other vaccine [108,109,110,111]. Another inactivated SARS-CoV vaccine study was conducted in BALB/c mice. The vaccine was prepared by culturing SARS-CoV in the Vero cell line followed by inactivation with β-propiolactone and purification by Sepharose 4FF column chromatography. The result demonstrated that a higher vaccine dosage was required to produce a higher neutralizing antibody titer [112].

Further observation declared that the vaccine works better if formulated with aluminum hydroxide as an adjuvant [112]. During the outbreak of another type of coronavirus in 2012, designated as MERS-CoV, more than 2468 cases occurred in 27 countries, with over 815 deaths occurring globally [113,114,115,116,117]. Two types of vaccine formulations were studied in mice, a spike protein and a whole inactivated MERS-CoV vaccine, where the latter presented better immune response in mice [118,119].

BBIBP-CorV is a whole virion inactivated vaccine manufactured by Sinopharm (Beijing, China) and formulated by inactivating the novel coronavirus strain HB02, isolated from a patient admitted to the hospital. The reason behind the selection was its replication efficiency in Vero cells [120,121]. CoronaVac, another inactivated SARS-CoV-2 whole virion vaccine manufactured by Sinovac Life Sciences Co., was assembled by propagating the virus in Vero cells, followed by the inactivation using β-propiolactone. Aluminum hydroxide was coupled to the vaccine formulation as an adjuvant [122]. During the phase 1/2 trial (human model; age 18–59 years), the production process of the vaccine was slightly different. The cell factory process was used to generate 50L Vero cell culture for the preclinical and Phase 1 trials, respectively, while for the Phase 2 trial, a bioreactor process was used for vaccine production [122].

Interestingly, the bioreactor production process was more appropriate, as the control of the environment was easier and accurate during vaccine production. Moreover, it was suitable for bulk production and ensured the biosafety requirements. The bioreactor batch of the vaccine contained a higher spike antigen than the vaccine used in the phase 1 trial, which was unexpected. Fortunately, it did not change the vaccine’s safety profile. Instead, it increased the immunogenicity of the vaccine candidate [122]. A separate study was conducted to evaluate the vaccine’s immunogenicity and safety in people aged 60 and older with CoronaVac vaccine. This study showed that the vaccine was suitable for older people as well [123]. A Phase 3 trial of CoronaVac was conducted in Turkey. It was a double-blind, randomized, placebo-controlled trial with 10 218 volunteers aged 18–59 years. Among the participants, nine cases of COVID-19 were seen in the vaccine group, whereas thirty-two cases were reported in the placebo group during a follow-up period of 43 days. The overall efficacy of the vaccine was 83.5% after the second dose. A total of 18.9% of the population in the vaccine group and 16.9% in the placebo group experienced minor adverse events; injection site pain was the most frequent adverse event [124].

BBV152 (Covaxin), another whole virion inactivated vaccine developed by Bharat Biotech (India), was produced by inactivating the virus and then formulating it with a toll-like receptor 7/8 agonist molecule, which was absorbed to alum (Algel-IMDG) [92]. The formulation of this vaccine was decided after its preclinical trial in BALB/c mice, New Zealand white rabbits, and the Syrian hamster model. Three types of formulations, BBV152A (0.3 μg Ag + Algel-IMDG), BBV152B (0.6 μg Ag + Algel-IMDG), and BBV152C (0.6 μg Ag + Algel), were assessed. The BBV152B formulation showed a 10-fold better immune response in mice, whereas BBV152A showed better results in the Syrian hamster. Confirmation of the safety and reactogenicity of the vaccine formulation enabled it to receive approval for trials in humans [125,126]. The other inactivated vaccine developed by Sinopharm (Beijing, China) (ChiCTP2000031809) was also made by isolating a SARS-CoV-2 strain WIV04 from a patient. The virus was cultivated in Vero cells, and the supernatant of the infected cell was treated with β-propiolactone to inactivate the virus. Alum was used as an adjuvant to the vaccine formulation [127].

After administering the inactivated vaccine formulations, mild side-effects such as pain in the injection site, fever, fatigue, headache, nausea, and vomiting were observed. Nevertheless, no profound negative result was reported, confirming that they were safe and immunogenic [92,120,122,123,127]. However, it is to be noted that, compared to the other vaccines, the adverse events of the BBV152 vaccine, developed by Bharat Biotech, were noticeably lower [92].

Human trials with the above-discussed inactivated vaccine formulations induced considerable immune responses. Seroconversion was reported in all participants of these vaccine studies. Overall, the efficacies of these inactivated vaccines, such as BBIBP-CorV (82.50%), BBV152 (81%), and CoronaVac (83.5%), were found to be almost similar to each other (Figure 3). The neutralizing antibody (Nab) titer was seen to be increasing with the increased dosage of the vaccine (Figure 4). However, the increased dosage caused unfavorable events in the trial population in each trial, as mentioned above, other than the CoronaVac trial (Figure 5) [92,120,127]. For CoronaVac, Nab titer and minor unusual reactions to the vaccine were the same in the higher (6 µg) and lower (3 µg) groups. Thus, 3 µg dose of the vaccine was selected for the Phase 3 trial [122]. The cellular and humoral response and T cell memory response were generated by BBV152, although the B cell memory response is yet to be assessed [92].

The trial of the BBIBP-CorV inactivated vaccine was limited by the absence of a longitudinal follow-up study and the assessment of the safety and immunogenicity in children. The vaccine trial was not designed to measure the efficacy of the vaccine [120]. The study of CoronaVac only evaluated humoral response in the Phase 2 trial and did not report immune response data on immunocompromised or more susceptible populations. The comparison of the Nab titer determined from the trial data with the titer observed in COVID-19 patients was not reported. Only the neutralizing antibody assay was undertaken in older people, which excluded T cell response observation. The study subjects of the aged group were healthy. Consequently, there were no assessment results about the safety and immunogenicity of the vaccine in immunocompromised people. The longitudinal follow-up result of the participants is yet to be observed [122,123].

The trial of BBV152 was conducted when the number of COVID-19 cases was rapidly increasing. However, the results of the trial did not confirm the evaluation of core vaccine efficacy. Besides, the study population lacked multi-ethnicity and longitudinal re-examination. Several participants reported minor unfortunate adverse reactions to the vaccine during the Phase 2 trial compared to the Phase 1 trial (Figure 5). However, the positive aspect was that it was designed with participants from diverse geographic sites and different age ranges. Regardless of their age, no differences were reported in the immune response. After completing the Phase 1 trial, 6 µg Ag + Algel-IMDG formulation was selected for the Phase 3 trial. Because of the current pandemic situation, it received approval for emergency use in India [92]. In the Phase 3 trial of BBV152, 24,419 participants enrolled. It was a double-blind, randomized, and multicenter trial. A total of 24 and 106 cases of COVID-19 were reported in the vaccine and placebo groups, respectively. Overall, the vaccine’s efficacy was 77.8%, whereas the vaccine’s efficacy against severe symptomatic and asymptomatic COVID-19 cases was 93.4% and 63.6%, respectively [128].

The trial design of the inactivated vaccine developed by Sinopharm did not include a plan for interim analysis in the original protocol. It was included while conducting the study, which was necessary for designing a Phase 3 trial. Despite producing a robust antibody response, whether the vaccine can provide protective immunity against COVID-19 is still unknown. The study result was based on a few groups of patients; hence, the study was unable to provide a comprehensive profile of suitability, immunity, and immune persistence [127]. These ongoing vaccine studies need further evaluation to assess if this vaccine provides long-lived immunity against SARS-CoV-2.

### 3.2. mRNA Vaccine

mRNA vaccine generally consists of the elements essential for the encoded protein to be expressed [129]. In the mRNA vaccine, 1-methyl-pseudouridine modification is incorporated in mRNA molecules, enhancing mRNA translation in the body [130]. The antigen is initially identified from the target pathogen. After sequencing and synthesizing, the gene is usually cloned into a plasmid. Before being delivered into the host, the mRNA is transcribed in vitro. After its injection into the body, it uses the host cellular machinery to translate the mRNA into the target antigen. Commonly, both humoral and cellular immunity are induced as the mRNA vaccine mimics the initial viral infection [131]. Chemokines and cytokines (i.e., IL-12, TNF) are produced at the injection site, generating robust innate immunity [132,133,134].

Compared to subunit, killed, live attenuated, and DNA-based vaccines, mRNA vaccines are preferred as they are safe and hardly have any harmful risk of infection. Besides, the mRNA vaccine is more stable, easily translatable, rapidly producible, and usually economical [135,136,137,138,139]. The easy availability of mRNA’s printing facility plays a crucial role in producing considerable quantities of mRNA that facilitate mRNA vaccine production [140]. The mRNA vaccine’s good adequacy and self-adjuvant properties elicit adaptive solid immune responses by releasing TNF-α, IFN-α, and other chemokines, by immune cells. Polypeptide and protein-based vaccines require additional adjuvants, whereas the mRNA vaccine does not require these. Again, mRNA vaccines express target proteins in the cytoplasm instead of entering the nucleus, making them more efficient than DNA vaccines [141].

In the early 1990s, mRNA vaccines were first identified; however, the production and stability complications did not allow them to advance into therapeutics in the coming years [142]. Later, around 2013, H10N8 and H7N9 mRNA vaccines against H10N8 influenza strain (A/Jiangxi-Donghu/346/2013) and H7N9 influenza strain (A/Anhui/1/2013) showed a robust humoral immune response in Phase 1 trials, whereby both the vaccines encoded hemagglutinin (HA) glycoprotein [143]. mRNA-based vaccines against HIV, CMV, Chikungunya, and Zika were developed before the COVID-19 pandemic [144]. However, none of these have been approved by the FDA to date. Due to advancements in nucleic acid technologies, their performance has improved in humans in recent years because of their new formulations and modifications [140]. During the current COVID-19 pandemic, Moderna, BioNTech, and Pfizer have launched mRNA-based vaccine candidates.

Moderna and Pfizer/BioNTech developed COVID-19 vaccines by implementing nanotechnology [145,146,147,148]. The human body has ribonucleases enzymes present in every tissue and ready to destroy scattered RNA. Ribonuclease creates a restricted environment for foreign RNA, which might originate from plants or animals. Moreover, negatively charged mRNA cannot cross the negatively charged cell membrane, making it challenging to restore mRNA integrity and enter the host cell. Researchers designed lipid nanoparticles to carry siRNA or mRNA into host cells. Lipid nanoparticles can be created to encapsulate RNA and shield it from ribonuclease. The lipid nanoparticles also allow passage through the cell membrane. Those nanoparticles can be decorated with ligands that allow targeting certain immune cell types.

Moreover, adjuvants can be added to the interior of the lipid nanoparticles for further upregulation of the immune system’s response to the vaccine. Lipid nanoparticle-based technology can be a beneficial solution to challenge RNA delivery to cells [149,150,151,152,153,154,155,156]. Vaccine development by nanotechnology improves nucleic acid delivery and conformation-stabilized subunit vaccines to lymph nodes. It triggers cellular and humoral immunity, preventing viral infection and disease severity [157].

The mRNA-1273 vaccine, manufactured by Moderna, encodes SARS-CoV-2′s Spike glycoprotein (S-2P antigen). An intact S1-S2 cleavage site and SARS-CoV-2 glycoprotein, along with the transmembrane, were anchored, making up the vaccine. Two successive proline substitutions at amino acid positions 986 and 987 stabilize S-2P on its prefusion conformation. These empowered the assurance of an atomic-level structure for the prefusion adaptation of spikes from endemic and pandemic strains counting HKU1, SARS-CoV, and MERS-CoV. Innovative structure-based vaccine design, modified nucleotides, and delivery methods by lipid nanoparticles are the principal reasons for mRNA-1273′s rapid immunogenicity.

The Phase 1 trial of mRNA-1273 on humans was conducted at the Kaiser Permanente Washington Health Research Institute in Seattle and the Emory University School of Medicine in Atlanta. After the first immunization, the day 29 enzyme-linked immunosorbent assay anti–S-2P antibody geometric mean titer (GMT) was found as 40,227 (25 μg group), 109,209 (100 μg group), and 213,526 (250 μg group), where a positive correlation was observed between dose and GMT. After the second immunization, day 57 GMT enhanced up to 299,751 (25 μg group), 782,719 (100 μg group), and 1,192,154 (250 μg group), respectively maintaining the initial response (Figure 4). However, approximately 21% of the 250 μg dose group had one or more severe unfortunate events (Figure 5) [158].

Significant immune responses to SARS-CoV-2 were reported in patients 18 years and older from the Phase 2 trials, which confirmed immunogenicity and safety of 50 and 100 μg of the mRNA-1273 vaccine. Within 28 days after the first vaccination, anti-SARS-CoV-2 spike binding and neutralizing antibodies were elicited by both doses of the mRNA-1273 vaccine. After the second vaccination, titers peaked by 14 days, which exceeded the COVID-19 patient’s convalescent sera level (Figure 4). Nevertheless, the Phase 2 trial had some limitations. The study population was not outlined at the time to be representative of those likely to get COVID-19. Statistical comparison of superiority or proportionality between doses was also not included in the study [159].

The Phase 3 trial showed 94.1% efficacy of the mRNA-1273 vaccine in preventing COVID-19 illness (Figure 3). The reactogenicity profile was similar to the Phase 1 data, and no unexpected concerns were detected. After the first dose of the vaccine, the level of reactogenicity was less than the zoster vaccine that was recently approved. The short duration of follow-up to investigate the safety and efficacy, and lack of any specified correlation of protection, were the drawbacks of the Phase 3 trial [160].

BioNTech and Pfizer manufactured both the BNT162b2 and BNT162b1, where a full-length spike of SARS-CoV-2 was encoded by BNT162b2. Two proline mutations were carried out to lock its prefusion conformation. As a result, this vaccine mimicked the intact virus and elicited immunity. On the other hand, the receptor-binding domain (RBD) of spike protein was encoded by BNT162b1. Trimerization was carried out by adding a T4 fibritin-foldon domain. Consequently, the immunogenicity was enhanced by the multivalent display. Lipids were used to formulate this vaccine and supplied as a buffered liquid solution [130,161].

The Phase 2/3 trial with the BNT162b1 vaccine was performed among 45 healthy adults, which assessed safety, tolerability, and immunogenicity of BNT162b1 at three dose levels (i.e., 10 µg, 30 µg, or 100 µg). BNT162b1 generated robust immunogenicity after vaccination, where dose levels had a positive correlation with the antibody titer. Moreover, the second dose increased SARS-CoV-2 neutralizing antibody titers and RBD-binding IgG concentrations. The second immunization with 100 μg was not administered because of the expanded reactogenicity and a need for seriously increased immunogenicity after a single dose compared with the 30 μg dosage. This trial proposed that dose levels between 10 µg and 30 µg with BNT162b1 might be well-tolerated and immunogenic [130,161].

However, several limitations were noticed. These included the kind, i.e., T cells versus B cells or both, and the level of immunity required to ensure protection from COVID-19 was unknown. In addition, the immune responses or safety were not assessed beyond two weeks after the 2nd vaccination. Again, as the study population was only up to 55 years old, the trial could not evaluate the plausible risk factors in the people beyond that particular age range [161,162].

A randomized, placebo-controlled, double-blind Phase 1 study of the BNT162b1 mRNA vaccine was conducted in younger and older Chinese adults to assess the preliminary safety and immunogenicity. This study showed an acceptable safety profile of BNT162b1 [163].

The safety and immunogenicity of three dose levels of BNT162b1 and BNT162b2 were also assessed in a Phase 1 trial on 145 healthy adults. In this trial, BNT162b2 showed less severity and incidence of adverse effects than BNT162b1 while eliciting a similar dose-dependent antibody titer, parallel to the GMT of SARS-CoV-2 convalescent patients, or even more in some cases. The reason behind the high reactogenic profile of BNT162b1 compared to BNT162b2 was the difference in their nucleotide sequences by which vaccine antigens were encoded and the overall RNA construct size. As a result, RNA molecules of BNT162b1 were five times higher than the same concentration of BNT162b2, which elicited high immune stimulation and a reactogenic profile. Besides, the lack of knowledge regarding how effective it would be in protecting COVID-19 in a real-world sense was a major limitation of the Phase 1 trial along with the assessment of humoral and cellular immunity in protecting COVID-19. Although the Phase 1 part of this trial evaluated several hypotheses, it was not large enough to allow systematic statistical comparisons and standardized among laboratories. Regarding all these issues, BNT162b2 was subsequently selected for Phase 2/3 safety and efficacy trials [130,161].

Phase 2/3 trials evaluated the safety, immunogenicity, and efficacy of BNT162b2 in preventing COVID-19. Two doses of 30 μg BNT162b2 conferred 95% effectiveness in preventing COVID-19 in patients aged 16 years of age or older and showed safety over a median of 2 months (Figure 3). However, this trial also had several limitations. Unexpected events such as right axillary lymphadenopathy, right leg paresthesia, paroxysmal ventricular arrhythmia, and shoulder injury were reported among BNT162b2 recipients (Figure 5) [94]. The vaccine also required freezing temperature for shipping and more extended storage.

A Phase 3 trial was also conducted to assess the safety, efficacy, and immunogenicity of BNT162b2 in 12- to 15-year-old participants. This study showed that a two-dose regimen of 30 μg of BNT162b2 was highly safe and immunogenic for adolescents (12 to 15 years of age) with an efficacy of 100% [164].

An exciting observation reported in BNT162b2 recipients were those who previously had SARS-CoV-2 infection; the anti-Spike titer increased approximately 140-fold within 19–29 days compared to those who had not been infected [165,166]. A single dose of this mRNA vaccine enhanced spike protein-specific antibody IgG level, ACE2 receptor binding inhibition reactions, and post-vaccine symptoms in individuals who were previously infected with SARS-CoV-2, similar to the second dose in individuals who were not infected [167,168]. Another study suggested that antibodies against SARS-CoV-2 nucleocapsid (N), RBD, and spike proteins (i.e., S1 and S2) were raised after the single dose of mRNA vaccine in those who were already seropositive or had a recent history of infection as compared to individuals with no history [169].

### 3.3. Viral Vector-Based Vaccine

In a viral vector-based vaccine, a gene/cDNA coding for a pathogen-derived antigen is incorporated into a non-pathogenic or attenuated viral species [170]. These non-pathogenic species serve as a vector. The recombinant vector immunizes against the pathogen while the gene product is expressed on its surface. For this purpose, a few sites are removed from the vector genome where the targeted pathogen’s foreign DNA can be integrated. After injecting, that foreign DNA can replicate within the host and express the following pathogen’s gene product, eliciting cell-mediated and humoral immunity. Different viruses, including adenovirus, vaccinia virus, canarypox virus, and attenuated poliovirus, act as viral vectors. Among them all, the adenovirus is relatively conventional in vaccine formulation [170,171].

Adenoviruses contain a distinctive icosahedral protein structure that encapsulates a linear double-stranded DNA genome of 36k base pairs. The viral genome has approximately a dozen capsid proteins without any lipid envelop and encodes nearly 35 proteins elicited in two phases, “early” and “late,” related to viral DNA replication [172]. Different types of adenoviral serotypes are separated from a variety of mammalian species. There are approximately 51 serotypes from humans, 27 from simians, and 7 from chimpanzees. Human serotypes are subdivided into six subgroups [173]. They are responsible for various clinical diseases that commonly infect the gastrointestinal and respiratory systems [174]. More than one serotype could be accountable for mild infection in most immunocompetent individuals, which is supposed to render lifelong immunity [172].

There are several distinctive features reflected by adenoviruses, making them a suitable option for vaccine formulation. Firstly, recombinant adenoviruses have a better safety record when used as a vector for gene therapy in humans. Because of its extensive tissue tropism ability, adenoviral vectors are supposed to exploit vaccine development against *M. tuberculosis* and influenza. Secondly, adenoviruses are highly immunogenic, driving up a strong and long-lasting immune response. For vaccine design purposes, human serotype 5 adenoviruses (AdHu5) are used broadly, while other non-human adenoviruses are also used in modern times. The adenoviral genome encodes five early proteins (E1a, E1b, E2, E3, and E4) responsible for DNA replication and evasion and a single late protein responsible for structural conformation. Removing the E1a gene (critical regulator of viral replication) from the adenoviral genome eliminates the virus’s ability to replicate, simultaneously increasing the potency to accommodate transgene cassettes up to 5000 bp in size. Deleting multiple genome units can augment the vector capacity for yielding multivalent vaccines against deadly pathogens [175].

Recombinant adenoviral vector vaccine trials were undertaken during the MERS-CoV epidemic. Dromedary camels were an animal reservoir of MERS-CoV. Two recombinant adenoviral vector vaccines, Ad5.MERS-S and Ad5.MERS-S1, were designed to generate an immune response against dromedary camels to eradicate MERS-CoV transmission from the reservoir to humans. Ad5.MERS-S encoded full-length S protein of MERS-CoV, and other Ad5.MERS-S1 encoded S protein’s S1 extracellular domain. Both vaccines immunized BALB/c mice and generated effective immunity [176].

BVRS-GamVac-Combi vaccine was manufactured by NF Gamaleya Research Institute of Epidemiology and Microbiology against MERS-CoV, where recombinant human adenoviruses 26 and 5 serotypes were used. Both vectors encoded the glycoprotein of the MERS-CoV. This vaccine immunized mice of C57BL/6 strain and common marmoset [177].

During the COVID-19 pandemic, the Beijing Institute of Biotechnology and CanSino Biologics manufactured the Ad5 vector COVID-19 vaccine. The replication-defective Ad5 vector vaccine encodes spike glycoprotein. Based on Wuhan-Hu-1 (GenBank accession number YP_009724390), an optimized full-length spike gene was cloned with tissue plasminogen activator signal peptide gene and E1 and E3 deleted Ad5 vector, resulting in an Ad5 vector COVID-19 vaccine developed utilizing the Admax system. It was manufactured as the liquid formulation in a vial containing 5 × 1010 viral particles/0.5 mL. The Phase 1 trial found that the vaccine is tolerable and immunogenic after 28 days of vaccination. From day 14 of immunization, rapid T cell responses were remarkable. Humoral immune responses peaked at day 28 of immunization. The Phase 2 trial showed a 5 × 1010 viral particles dose as safe for vaccination. Significant immune responses were induced after a single immunization. The short duration of follow-up, small cohort size, and absence of randomized control groups limited the findings from the Phase 1 trial. The study was also not statistically powered to assess the level of any side-effects. The Phase 2 trial, too, also presented some limitations. The problems began as the study started before complete data analysis from the Phase 1 study was accessible. Consequently, it failed to show the difference between different dosage groups. Children were also not included in this trial [178,179].

The Institute of Biotechnology, Academy of Military Medical Sciences, PLA of China, developed an aerosolized adenovirus type-5 vector-based COVID-19 vaccine (Ad5-nCoV). The Phase 1 trial was performed at Zhongnan Hospital to evaluate the vaccine’s safety, tolerability, and immunogenicity administered via inhalation. This study showed that aerosolized Ad5-nCoV is painless, simple, well-tolerable, and an aerosolized dose-induced antibody and cellular immune responses are equal to a fifth of the usual injected dose [180].

Among several candidate vaccines, the three most promising adenoviral vector-based vaccines have been going through clinical trials to ensure the host’s safety, efficacy, and immunogenicity while inducing an immune response. Adenoviral vector combined with DNA and poxviral vector to induce immunogenicity showed cellular and humoral response enhancement. The Oxford/AstraZeneca vaccine contains a homologous adenoviral vector that could mitigate the efficiency of the second dose due to anti-vector immunity. The “chimpanzee adenovirus” was an excellent vector for vaccine formulation from previous vaccine preparations against MERS-CoV. The Phase 1 clinical trial showed promising results, maintaining potency and protecting non-human primates against MERS-CoV. ChAdOx1 MERS exhibited safety and efficacy, which expressed cellular and humoral immune responses at the highest dose (5 × 10^9^ viral particles).

The Oxford/AstraZeneca vaccine is an adenoviral vector-based vaccine comprising DNA that encodes surface glycoprotein protein embedded in a capsid from a modified chimpanzee adenovirus, ChAdOx1 (replication-deficient simian adenovirus vector). A single vaccination with ChAdOx1 nCoV-19 into rhesus macaques showed potent humoral and cellular responses in a preclinical trial stage. A high-dose vaccination into a non-human primate showed protection against lower respiratory tract infections. After the Phase 1/2 clinical trial, the ChadOx1 nCoV-19 vaccine showed anti-spike IgG response, early T-cell response, and neutralizing antibody response, which illustrated an admissible safety profile with an enhanced humoral and cellular response. Vaccinated participants presented different amounts of immune cell clusters of B-cells, T-cells, and NK cells. Strong activated B-cells are found, and anti-spike IgG and IgA antibodies against SARS-CoV-2 spike protein were identified from vaccinated individuals. Identical CD4+ and CD8+ T-cell patterns were observed, responsible for the expression of CD69 and Ki-67, between days at 7 and 28 after vaccination. Production of cytokines (TNF-α and IFN-γ) by CD4+ T-cells was also identified on day 14 after immunization. On the other hand, NK cells expressed cytotoxic activity against viral infection at the highest rate at day 28 [93,181].

A published report showed that a booster dose is more efficient in inducing multifunctional antibody titers with different types of effector mechanisms, including antibody-dependent neutrophil/monocyte phagocytosis, natural killer (NK) cell degranulation, complement activation, and cellular phagocytosis against SARS-CoV-2 infection. Strong T-cell responses have also been delineated, in which highly activated cytotoxic T-cells destroy virus-infected cells to stop the further cell-to-cell spread of the virus. On the other hand, helper T-cells play a supporting role in activating B-cells for frequent antibody production. The booster dose is less reactogenic compared with the priming dose. Local and systemic reactions were consistently reduced after the second dose (Figure 5). A booster dose necessitates its importance in the restoration of sustainable immunity [182].

The Phase 3 trial of the ChadOx1 nCoV-19 vaccine has shown improved findings, with the vaccine being more well-tolerated and efficient (70.4%) in older than younger adults and similarly immunogenic for all age groups [183]. A well-accepted safety and efficacy profile of AZD1222 (ChadOx1 nCoV-19) has been established after analyzing ongoing trials in Brazil, South Africa, and the UK, making it an efficient and robust candidate vaccine against SARS-CoV-2 globally (Figure 3 and Figure 5) [184]. Epidemiologic efficacy of the Oxford/AstraZeneca ChAdOx1 nCoV-19 vaccine against the B.1.351 variant was 10.4% (95% CI, −76.8 to 54.8). However, the vaccine effectiveness against the B.1.1.7 variant was comparatively higher than the exhibited efficacy of 70.4% (95% CI 43.6–84.5) for B.1.1.7 and 81.5% (67.9–89.4) for non-B.1.1.7 lineages. However, the neutralization activity of ChAdOx1 nCoV-1 decreased in the B.1.1.7 variant compared to the non-B.1.1.7 variant [185,186].

Mutations in the RBD and N-terminal domain (NTD) of the SARS-CoV-2 spike gene are the major concern for vaccine development in recent pandemics [186]. RBD mutations (N501Y mutation) are responsible for increasing affinity to the ACE-2 receptor. On the other hand, E484K and K417N RBD mutations and mutations in NTD are accountable for escaping from the neutralizing antibody response. The B.1.1.7 lineage, first identified in the UK, contains the N501Y mutation with 53% increased transmissibility. Another mutated clan, named B.1.351, identified in South Africa, includes three RBD mutations and five NTD mutations. An independent lineage found in Brazil also adopted E484K, K417N, and some B.1.351 mutations [186].

An individual analysis of the Oxford/AstraZeneca vaccine against B.1.351 (South African variant) was undertaken between 24 June and 9 November 2020, where 2026 participants were enrolled. The T-cell response was not effective. Thereby, significant portions of viral antigens of B.1.351 variants remained flawlessly active. Furthermore, the vaccine failed to show a protective immune response against the B.1.351 variant, whereby the vaccine efficacy against the variant was 10.4% (95% CI, −76.8 to 54.8) [186].

Sputnik V, a heterologous adenoviral vector-based vaccine manufactured by Gamaleya Research Institute, was designed with two recombinant adenovirus vectors, adenovirus type 26 (rAd26) and adenovirus type 5 (rAd5), and both contain full-length glycoprotein S [187]. rAd26 vector was previously used for different vaccine candidates, such as Ad26.ZE.BOV against Ebola virus; Ad26.Mos.HIV, Ad26.Mos4.HIV, and Ad26.ENVA.01 against HIV; Ad26.CS.01 against Malaria; and Ad26.ZIKV.001 against Zika. These candidate vaccines are being tested, and clinical studies are still ongoing [188,189,190,191,192].

Sputnik V vaccine was developed in two versions, i.e., frozen (Gam-COVID-Vac) and lyophilized (Gam-COVID-Vac-Lyo). The Gam-COVID-Vac study took place at the branch of Burdenko Hospital, and the volunteers who took part in this study were military personnel and civilians. The study of Gam-COVID-Vac-Lyo was undertaken at Sechenow University, and all volunteers were civilians [187]. The vaccine showed strong immune response and protection in non-human primates against SARS-CoV-2 and displayed 100% defensive measure against a lethal version of SARS-CoV-2 in a preclinical study with immunosuppressed hamsters.

After the Phase 1/2 trials, Sputnik V showed T cell responses in healthy adults and decent titers of neutralizing antibodies. Collective data exhibit higher immunogenicity with robust cellular and humoral immune responses, resulting in higher antibody titers in vaccinated participants than the individuals with convalescent plasma [187].

The Phase 3 trial took place at 25 hospitals and polyclinics in Russia. The Gam-COVID-Vac trial showed 91.6% efficacy against COVID-19 and 100% protection against severe COVID-19 (Figure 3). In this trial, people 60 years or older were given importance in the vaccine-inducing immune response’s protection measures and efficiency. Results showed the ability to induce a virus-neutralizing humoral reaction in 60-year-old individuals. Vaccine efficacy did not alter significantly in young adults and old-age vaccinated participants. However, all risk groups, including children and pregnant women, were not enrolled in the Phase 3 trial. The vaccine is developed in liquid form (stored at −18 °C), and the freeze-dried (held at 2–8 °C) formulation is helpful in the distribution of vaccines in different weather conditions globally. The Phase 3 clinical trial showed a compatible safety profile and robust immune responses in all age groups from young to old participants (Figure 4 and Figure 5) [193]. The Phase 3 trial showed the sputnik vaccine required two doses to reach 91.6% efficacy, with a 79.4% efficacy after one dose as emergency administration [187,193].

FINLAY-FR-1A, a recombinant dimeric RBD-based vaccine manufactured by a Cuban epidemiological research institute, Finlay Institute, against COVID-19, showed a safe and reactogenic outcome in the Phase 1 trial (Figure 5). Secondary outcomes evaluated vaccine immunogenicity. One week after vaccination with a single dose, antibody response enhanced more than 20-fold compared to the Cuban convalescent serum panel’s median [194].

VXA-CoV2-1, developed by Vaxart, is an attractive recombinant vaccine candidate against SARS-CoV-2 and is an oral vaccine formulation. The preclinical trial of this vaccine was conducted in 6- to 8-week-old female Balb/c mice. Due to the mice’s inability to swallow pills, they were injected with 20 µL of the vaccine formulation. Two types of recombinant vaccine formulations, rAd-S-N (vector that expresses S and N protein) and rAd-S1-N (vector that expresses a fusion protein of S1 domain and N protein), were assessed. The former was selected for GMP manufacturing, as it induced a higher immune response, including vaccine-induced T cell response and the production of IFN-γ, TNF-α, and IL-2 CD4+ T cells. This vaccine candidate presents several advantages. Because it is an oral formulation, this makes it more accessible, as it is easier to administer. It is also easier to store, being an oral tablet vaccine, eliminating the need for cold storage transport and holding it in a refrigerator once delivered. Another advantage of this vaccine candidate is it is safe [195].

Another promising candidate vaccine is Ad26.COV2.S, manufactured by Janssen Pharmaceuticals. Ad26.COV2.S utilizes a recombinant, replication-deficient adenovirus serotype 26 (Ad26) vector encoding a stabilized SARS-CoV-2 spike (S) protein educed from the first clinical isolate of the Wuhan strain (Wuhan 2019; whole-genome sequence, NC_045512). Ad26 vector-based vaccines are usually safe and highly efficient and are also being used in the Sputnik V vaccine [187,196].

The Ad26.COV2.S vaccine has been tested in adult and aged rhesus macaques dividing into one- and two-dose regimens to evaluate the protective immune response and efficacy. A two-dose Ad26.COV2.S regimen promoted an ascending neutralizing antibody response compared to a single-dose regimen. However, neutralizing antibody responses were stable for a minimum of 14 weeks in one-dose regimens of the Ad26.COV2.S vaccine, and it also upregulated the humoral immunity and Th1 cellular responses in aged NHP [197].

Eight hundred five healthy adults have been assigned for participating in a Phase 1-2a trial of the Ad26.COV2.S vaccine. In the trial, cohort 1 belongs to participants aged 18–55 years, and cohort 3 belongs to 65 years or older-aged participants. A single dose of the Ad26.COV2.S vaccine elicited both neutralizing antibody and spike-binding antibody responses in 90% of participants on day 29 and reached 100% at day 57 with an increase in the titer. In addition, the CD4+ T-cell responses were found in 76–83% of participants in cohort 1 and 60–67% of those in cohort 3. On the other side, strong CD8+ T-cell responses were detected in all participants but at a comparatively lower level in older individuals than in younger [196].

The Phase 3 trial demonstrated the efficacy of a single dose of the Ad26.CoV2.S vaccine. A total of 67% and 66% efficacy was shown in participants against moderate to severe–critical COVID-19 disease with an onset at least 14 and 28 days after vaccine administration, respectively. The vaccine efficacy was 76.7% and 85.4% for severe-critical COVID-19 with onset at days 14 and 28, respectively. Reactogenicity was higher with the vaccine group, but a casualty of adverse events was not serious in the vaccine group. Overall, a single dose of Ad26.COV2.S is protective against symptomatic and asymptomatic SARS-CoV-2 infections and effective against severe/critical disease to reduce hospitalization and death. The Janssen adenovirus virus vaccine against B.1.351 variant has shown a 57% efficacy [198].

### 3.4. Nanoparticle-Based Peptide Vaccine

Nanotechnology has played an influential role in vaccine development with variations based on nanoparticles’ different compositions, shapes, sizes, and surface properties. Nanoparticles, being smaller in size, can quickly enter into living cells through endocytosis [80,199]. Different types of nanoparticles are being used in vaccine development, including polymeric nanoparticles, inorganic nanoparticles, liposomes, immune-stimulating complex (ISCOM), virus-like particles, self-assembled proteins, and emulsions [199,200,201]. Nanoparticles are most commonly used as immunostimulants or delivery materials. In vaccine formulation, the association between nanoparticles and antigens is essential. Nanoparticles act as a temporary carrier and protector of the antigen, which needs to reach the desired location. By interacting with the antigen, nanoparticles enhance immunogenicity and antigen processing, which activate immune responsive pathways [200,201].

Nanotechnology-based vaccine mechanisms are highly efficient, whereas solid nanocarriers transport the core antigen portion of vaccines into the gut-associated lymphoid tissues and mucosa-associated lymphoid tissues, ensuring proper delivery through oral or mucosal routes. Core particles are taken up by the dendritic cells and macrophages, which improve the cellular uptake of antigens and upregulate the antigen recognition and presentation. Nanoparticles are coated with immune cell-targeting molecules that bind with the cellular receptors to stimulate the specific and appropriate immune response [72,77,201,202,203,204,205]. However, no contextual and relevant results have yet been published regarding nanotechnology-based vaccines since the outbreak of severe acute respiratory syndrome (SARS-CoV) and middle-east respiratory syndrome (MERS-CoV) other than the current COVID-19 pandemic [206].

In this ongoing SARS-CoV-2 pandemic, a subunit vaccine (NVX-CoV2373) has been developed using full-length glycoprotein S and administered with Matrix-M adjuvant into non-human primates and mice models, spurring Th1-dependent B- and T-cell responses, production of hACE2 receptor blocking antibodies, and SARS-CoV-2 neutralizing antibodies. No vaccine-related adverse effects were reported in mice models, which encouraged further clinical development of NVX-CoV2373 against COVID-19 [207].

Researchers fabricated a modified “spike gene” of SARS-CoV-2 and installed it into baculovirus, which can only infect insects. Hence, selective moth cells were chosen and infected with the recombinant baculovirus. Consequently, the infected cell started to produce spike proteins assembled to form full-length spike protein similar to SARS-CoV-2. After that, spike proteins were purified and fixed with nanoparticles, which were used as a vaccine. Before being mixed with adjuvant distilled from soapbark plants, this vaccine attracted the immune cells to the injection site and activated the solid immune response to nanoparticles. Antigen-presenting cells (APC) uptake and present the spike nanoparticles on its membrane to T lymphocytes via major histocompatibility complex (MHC). T lymphocytes activate the antibody-producing B cells. A different type can be started by APC, called a killer T cell, which can recognize coronavirus-infected cells and destroy them before the further proliferation of new viruses [207,208].

In Phase 1/2 trials, 131 healthy adults of different age groups participated in two-dose regimens and were administered with 5 μg and 25 μg of rSARS-CoV-2 with or without the Matrix-M1 adjuvant. Considerable safety results and the ability to induce immune responses with higher amounts of neutralizing antibodies were found in groups with adjuvant compared to groups without adjuvant. After the second vaccination with a similar dosage, the antibody response in participants had surpassed compared to the convalescent serum from COVID-19 patients. This report expresses the advantage of Matrix-M1 in the case of accelerating the functional antibody and T-cell response. No serious local or systemic adverse events occurred with the vaccinated groups. Body pain, joint pain, and fatigue were the most common systemic events that have been reported after the Phase 1/2 trial (Figure 5) [209].

A total of 15,187 participants were included in the Phase 3 trial of NVX-CoV2373, which was found to be 89.7% effective against both B.1.1.7 and non- B.1.1.7 variants. This B.1.1.7 variant is more transmissible and infectious than the previous strains. The vaccine efficacy of NVX-CoV2373 was higher than that of the ChAdOx1 nCoV-19 vaccine (Oxford/AstraZeneca) (70.4%) after the Phase 2-3 trial. The NVX-CoV2373 vaccine exhibited a lower efficacy level (51.0%) against the B.1.351 variant [185,208] (Figure 3). Scientists suggested that the NovaVax can be effective for a long time and prevent future coronavirus infections. It is easy to store for a long time at 4 °C, making it convenient to transport [208].

## 4. Efficiency of Vaccines Observed after Phase 3 Trial

The efficiency of some of these vaccines was assessed in participants with various conditions. The Phase 4 trial of CoronaVac included participants with autoimmune rheumatoid arthritis (ARD) and a healthy control group. The outcome showed that anti-SARS-CoV-2 IgG seroconversion (SC) and neutralizing antibody (NAb) positivity was reduced six weeks after the second dose by ≥15%. The SC and Nab response was lower in the ARD group than in the control group [210].

In a study of CoronaVac, it was seen that the vaccine is capable of inducing immunogenicity in 63.8% of the study population who have cancer (solid organ tumor) and 59.5% of the people with cancer and also receiving at least one cytotoxic chemotherapy [211]. Lower antibody responses were found in another study of participants aged 65 years and older, and 51.1% of the total study population had at least one comorbid disease. There was a significant difference in the mean antibody titer of participants with at least one comorbid disease. Mean antibody levels decreased with decreasing age and comorbid disease [212]. The vaccine could not produce detectable antibody responses in hospital workers with immune-mediated disease (IMD) and people aged 65 years and older. Participants who were taking immunosuppressive drugs were significantly less likely to produce antibodies [213]. However, another study compared antibody response to the vaccine in previously infected and uninfected healthcare workers. The outcome suggests that the vaccine is capable of eliciting higher antibody responses in previously infected participants [214]. In a case–control study of CoronaVac during an epidemic of COVID-19 associated with the gamma variant in Brazil, it was seen that the vaccine reduced hospital admission and death in participants aged ≥70 years. Vaccine-induced protection was higher in participants who completed two doses, and the vaccine’s efficacy declined with increasing age [215]. Another study concluded that CoronaVac produced a lower neutralizing antibody response against alpha and gamma variant than D614G variant in participants who received two vaccine doses [216].

In an interesting observational case study in Bahrain, a family showed that two participants who had taken two different vaccines were in contact with COVID-19 patients who did not show any symptoms. As a response to the study, the country started giving boosters of the Pfizer vaccine to Sinopharm vaccine recipients. It concluded that the Sinopharm vaccine does not prevent people from getting infected [217]. Another study of the Sinopharm/BBIBP-CorV vaccine showed that the vaccine could induce an antibody response in 95% of participants, although the seroconversion rate was lower in older individuals (>60 years). The vaccine generates the same level of antibody response against B.1.617.2, B.1.351, and ACE-2 receptors, as seen in the case of natural infection [218].

Although the antibody titer produced by BBV152/Covaxin is low for the annulment of beta and delta variants, its neutralization potential is well established; 97.8, 29.6, and 21.2 GMT titers were found in recovered cases of B1, Beta, and Delta variants [219]. Another study demonstrated that Covaxin was able to induce protection against the B.1.1.7 variant [220].

An observational study was conducted in Israel residents aged 16 years and older following a nationwide vaccination campaign using national surveillance data to assess the real-world effectiveness of two doses of BNT162b2 against various SARS-CoV-2 outcomes and evaluate the public health impact. The effectiveness of the vaccine was 95.3% against SARS-CoV-2 infection, 91.5% against symptomatic COVID-19, 97.2% against COVID-19-related hospitalization, and 96.7% against COVID-19-related death at 7 days or longer after the second dose, where the B.1.1.7 SARS-CoV-2 variant was the dominant strain [221].

Another population study in Qatar showed that the mRNA-1273 vaccine is highly effective against infection induced by B.1.1.7 (Alpha) and B.1.351 (Beta) variants. The vaccine’s effectiveness against B.1.1.7 and B.1.351 variants was 81.6% and 95.7% after the first and second dose [222].

A case–control study was conducted in the population of Qatar to assess the real-world effectiveness of the BNT162b2 (Pfizer-BioNTech) and mRNA-1273 (Moderna) vaccines against the Delta (B.1.617.2) variant. At ≥14 days after the second dose, effectiveness against the Delta variant was 89.7% for BNT162b2 and 100.0% for mRNA-1273 [223].

Another study showed that the BNT162b2 COVID-19 vaccine elicits robust SARS-CoV-2-S antibody and T cell responses in nursing home residents (NHR) Clínico-Malvarrosa Health Department, Valencia (Spain). In SARS-CoV-2 naïve NHR, the seroconversion rate was 95.3%, similar to controls (94.4%) [224].

A further study among hemodialysis patients observed that poor immunogenicity was generated at 28 days after a single dose of BNT162b2. Consequently, the second dose should be administered without any delay in this population [225]. Further research has also suggested that this vaccine is safe for youths and young adults having a previous history of acute lymphoblastic leukemia and allergy to PEG-asparaginase [226].

## 5. Overall Comparison of Vaccine Candidates in the Trial Phase

According to the studies, all the reported vaccine candidates based on inactivated, mRNA, recombinant, and nanoparticle-coupled strategies showed promising efficacy (Figure 3). The seroconversion and the neutralizing antibody titers were observed in almost every trial for each vaccine candidate, where the seroconversion mainly started from days 7–14. In nearly every Phase 2 and 3 trial, the overall seroconversion rate was approximately 80–100% (Figure 4). However, one or more unwanted events comprising both the systemic and local systems were reported in every phase of trials for all registered vaccine candidates (Figure 5). Among mRNA-based vaccine candidates, the mRNA-1273 vaccine was reported to have a higher rate of unpleasant events than the other. ChadOx and recombinant Ad5 assisted in developing more frequent side effects in recombinant vaccines than Sputnik V and FINLAY-FR-1A. The nanoparticle-based vaccine candidate showed numerous immunogenic events (>80%) (Figure 5).

## 6. Gender-Based Adverse Events of COVID-19 Vaccines

Some concerns have been raised about the safety of the COVID-19 vaccines. For example, it was suggested that the vaccines could have an impact on pregnancy and damage fertility. In particular, mRNA vaccines were claimed to be cross-reactive with the human placental protein syncytin, potentially causing placental damage. However, these were unfounded speculations. Vaccines are completely safe during pregnancy and provide excellent protection for the baby. Pregnant women should be urged to acquire COVID-19 vaccines because they are also at risk of becoming infected during pregnancy [144,227,228,229]. During trials, pregnant people were excluded, and participants were strictly asked to avoid getting pregnant. However, 57 pregnancies were found across the trials of the three most highlighted vaccines that have been approved in the UK. After completing the trials, 0% miscarriage results were found in both the Moderna and Pfizer/BioNTech vaccines, and 2% accidental miscarriages were found in patients receiving the Oxford/AstraZeneca vaccine, which is small compared to control groups. This report indicates that vaccines do not have any harmful effects on early pregnancy and do not restrain pregnancy in humans [227,230,231,232].

After vaccination, adverse events (AEs) have been reported across all age groups above 18 years old. However, adults aged 65 and above reported more serious AEs (SAEs) when compared with the 18 to 64 years old group, which included deaths and dyspnea. In contrast, most of the AEs reported by the 18 to 64 age group were not serious in nature. Moreover, it has been noted that, when compared to females, males are more likely to experience SAEs, while females are more likely to report AEs compared to males. The majority of the AEs were recorded within one week of the initial dose. Overall, the vaccine developed by Pfizer and BioNTech has resulted in more AEs. Approximately 10% of the reports were serious, with 2% of them involving death, with the majority being in patients aged 65 years and above. Additionally, among the 20% of the subjects who visited the ER, more than 5% were involved with hospitalization, while 10% were involved with office visits. Headache, fatigue, dizziness, chills, pyrexia, nausea, pain, injection site pain, and pyrexia were among the top ten non-serious AEs following COVID-19 immunization [233].

In another study, thirty-two thousand forty-four subjects from VigiBase were assessed to evaluate SAEs associated with various COVID-19 vaccines based on age, gender, and the severity of the adverse events. Among these subjects, 80% were females. A total of 103,954 adverse events, at a rate of 3.24 AEs per subject, were reported. Furthermore, among the total AEs reported, 28,799 (27.7%) AEs from 8007 subjects were recorded as SAEs. The majority of SAEs were seen in Europe (83%) among females aged between 18 and 64 years (80.74%). Vaccination with BNT162b2 (Pfizer) was linked to the majority of SAEs (74%) recorded. General illnesses (30%) were the most common SAEs, followed by the nervous system (19.1%) and musculoskeletal (11.2%) disorders on a system-wise classification. Individually, headache (8.1%) was the most prevalent ailment, followed by pyrexia (7%) and fatigue (5.1%). According to a clinical trial linked to the Moderna vaccine, SAEs accounted for 26.73 percent of all AEs recorded in the VigiBase, with death occurring in 1.23% of all SAEs. As per the analysis, myocardial infarction (0.03%), cholecystitis (0.02%), and nephrolithiasis (0.02%) were the most prevalent SAEs reported. The number of serious adverse events (SAEs) reported with the various vaccines was lower than the non-serious ones, and the death rate was low among all vaccines. Overall, female vaccine recipients reported more local symptoms, such as injection site discomfort, redness, and swelling, as well as certain systemic events, such as joint pain, myalgia, headache, back pain, abdomen pain, fever, chills, and hypersensitivity reactions, than male vaccine recipients [234].

During vaccination with ChAdOx1 nCoV-19, a minority of the participants (*n* = 13) reported thrombosis as an AE, which led to the halting of the vaccination program. Sinus and/or cerebral vein thrombosis were noticed in those individuals after the vaccination. However, prior to the vaccination, these individuals were preoccupied with thrombocytopenia, which would have probably led to the immunological cascade eventually causing thrombosis [235,236]. According to the Society of Thrombosis and Haemostasis Research (GTH), such a phenomenon may occur due to the induction of antibodies after vaccination, which may react against platelet antigens, ultimately creating a massive activation of platelets via the Fc receptor that either may be independent or dependent on heparin. However, it is to be noted that the vaccination program was restarted due to a lack of strong correlation between the ChAdOx1 nCoV-19 vaccine and thrombosis. Furthermore, it was also stated that individuals are more unlikely to develop thrombosis after vaccination, though several guidelines have been put in place if any such untoward major incident may arise [236].

## 7. Importance of the Integration of Diagnostic Assays Pre- and Post-Vaccination

Phase 3 trials measure the efficacy of a vaccine by assessing how well it works based on double-blinded placebo-controlled trials [67,237,238]. The efficacy of vaccines is based on patients’ susceptibility and chances of developing infection [239]. RT-PCR is used to detect reinfection cases after vaccination. Mass testing with highly sensitive rapid antigen testing (RAT), capable of detecting COVID-19 infection in the acute phase, is crucial in monitoring reinfection cases post-vaccination [240,241,242]. Vaccine antigen-specific sero-monitoring and memory T/B-cell persistence are also essential determinants for the requirement of booster doses and epitope modification in vaccine designing.

Another method for determining vaccine efficiency is serological testing. Seroconversion after vaccination indicates that protective immunity has been activated in the vaccinated host [243]. The rapid diagnostic test (RDT), enzyme-linked immunosorbent assay (ELISA), chemiluminescence immunoassay (CLIA), and neutralization assay are the different types of serodiagnostic tests that can be employed to identify seroconversion, persistence, and longitudinal dynamics of both the neutralizing and non-neutralizing antibodies [186,244,245,246,247,248,249,250,251]. Besides, serological testing is crucial for immuno-compromised or autoimmune patients, who generally have altered immune responses [252,253,254]. However, activation of cell-mediated immunity (CMI) without humoral immunity post-vaccination in a few cases elucidates the importance of assays detecting both cell-mediated immunities as well as humoral immunity in testing the activation and waning effects of a vaccine. Thus, integrating diagnostic assays with the vaccination program helps identify the vulnerable population who requires booster doses and assesses vaccines of different strategies for their requirements and frequencies of booster shots.

## 8. Mix-and-Match Approach

The worldwide distribution, proper management, and shortages of vaccines of the same brand in different countries are a concern. To address this and potentially enhance responses, tactics including the “mix-and-match” approach are under trial. Here, the heterologous vaccination of two different types of vaccines is administered as prime and booster doses, respectively. This strategy is not entirely a novel idea, as such heterologous vaccination methods have been applied previously for HIV and Ebola [255]. In the case of the SARS-CoV-2 vaccine, a few heterologous approaches have already been studied in different countries. Heterologous combinations such as recombinant (ChAdOx1 nCoV-19) and mRNA (BNT162b2), inactivated (CoronaVac), and recombinant (ChAdOx1 nCoV-19), as well as inactivated (BBIBP) and mRNA (LPP-spike-mRNA), have been studied. To date, a significant augmentation of anti-spike antibody memory B and T cell responses with a few mild side effects, including headache, chills, and fatigue, have been observed [256,257,258,259,260]. Currently, a thorough evaluation of the homologous and heterologous prime booster of COVID-19 vaccines is underway.

## 9. Genomic Surveillance and Vaccine Up-Gradation

Many people were initially reluctant to be vaccinated as COVID-19 vaccines were rapidly developed [261,262]. Ideally, researchers aim to develop a single dose-based vaccine that is highly effective. However, it is challenging, as different types of vaccines present different immunoreactivity pathways based on host factors such as persistence of protection, co-morbidities, and ethnicity. Furthermore, the efficacy of different types of vaccines in providing protection depends on the emerging variants of SARS-CoV-2, some of which can evade the immune system. Previously, E848K escape mutation of SARS-CoV-2 and other mutations were reported in different countries through the next-generation sequencing (NGS) method [263,264,265,266,267]. The recent concern about reinfection by the new lineage P.1 (alias of B.1.1.28.1) of SARS-CoV-2, similar to the lineage B.1.1.28, evolved in Brazil or Bengal variant in West Bengal B1.1.618, is a puzzling phenomenon for the vaccination process [268]. Inclusion of several mutations of an amino acids (i.e., S: E484K, S: K417T, and S: N501Y) might work as the means to address the question of inadequacy of vaccines in providing extended protection in the case of immune escape mutations [269]. Vaccine manufacturers and researchers should carefully monitor the emerging variants and tweak their vaccines accordingly, addressing immune escaping mutations. Furthermore, a local genomic knowledge-driven vaccination policy could significantly improve the fight against COVID-19.

## 10. Conclusions and Future Directions

We have found that vaccines developed using mRNA technology show overall better efficacy than the other strategies. However, in general, conventional inactivated vaccines show less frequent side effects, but interestingly, all vaccines exhibit a similar level of humoral immunity. The vaccine manufacturers should be careful about escape mutation, reinfection, single-dose efficacy, and minimizing unusual events while keeping the effectiveness stable or enhanced. Researchers collaborating with manufacturers should undertake studies assessing vaccine efficacy when vaccines of the same technology from different manufacturers are administered in successive doses. Consequently, they should check whether they can complement each other. Vaccine manufacturers should also systematically modify vaccines with escape mutations. Moreover, trials should also be performed with mixing or subsequent administration of vaccines of different technologies to check whether these can give a broad range of cross-protection against current emerging variants. Besides, the vaccine trials should continually be enrolled in other regions, ages, ethnicities, and health conditions. Proper funding, rigorous research, and thorough analyses are required to overcome this situation in no time.

## 11. Limitation of the Study

Due to the specific search strategy restriction for this systematic review, we could not include any study published after 1 September 2021. In addition, since we searched only online databases (i.e., PubMed, ScienceDirect, and Google Scholar) for original published articles, we could not consider any data or reports available on random online websites other than two reports from the Google Scholar database discussing the efficacy of BBIBP-CorV (Table 2) [270,271].

## 12. Article Highlights

(1)This review was selected through an appropriate systematic search strategy.(2)This review discusses different types of vaccines strategies (i.e., inactivated, mRNA based, recombinant, and nanoparticle-based vaccines) developed so far for SARS-CoV-2.(3)Vaccines from a variety of manufacturers and countries have been discussed and categorized separately, based on their types.(4)The overall efficacy and safety of each of the candidates based on trial has been discussed.(5)The limitations of the clinical trials, issues, and other perspectives have been discussed.

## Figures and Tables

**Figure 1 vaccines-09-01387-f001:**
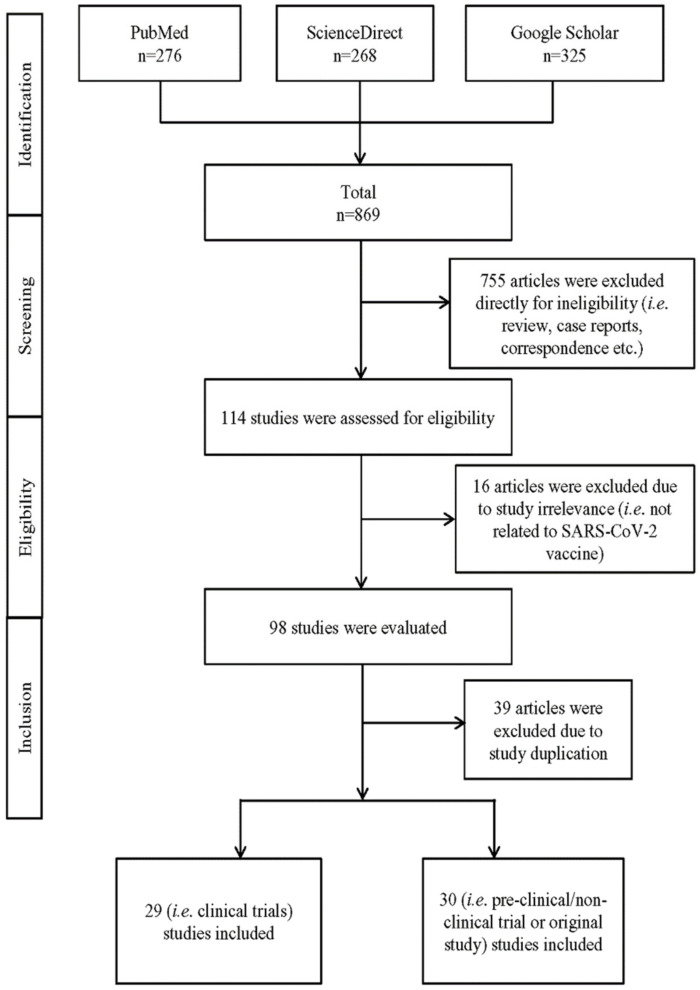
A simplified PRISMA diagram of methodology. Primarily, a total of 869 articles were identified by our search strategy from different online databases (i.e., PubMed, ScienceDirect, and Google Scholar). From this, 755 articles were subsequently excluded due to ineligibility as they were case reports, review articles, correspondence, letters, or articles other than the original full-length article. From the remaining 114 articles, 16 articles were subsequently excluded as they did not match our study criteria (i.e., not related to SARS-CoV-2 vaccines). Ultimately, 59 articles were included for this systematic review after excluding duplicate articles (*n* = 39). Among the final 59 studies, 29 were clinical trials, and 30 were pre-clinical, non-clinical, or other original studies on vaccines. Quality assessments were undertaken for clinical trials.

**Figure 2 vaccines-09-01387-f002:**
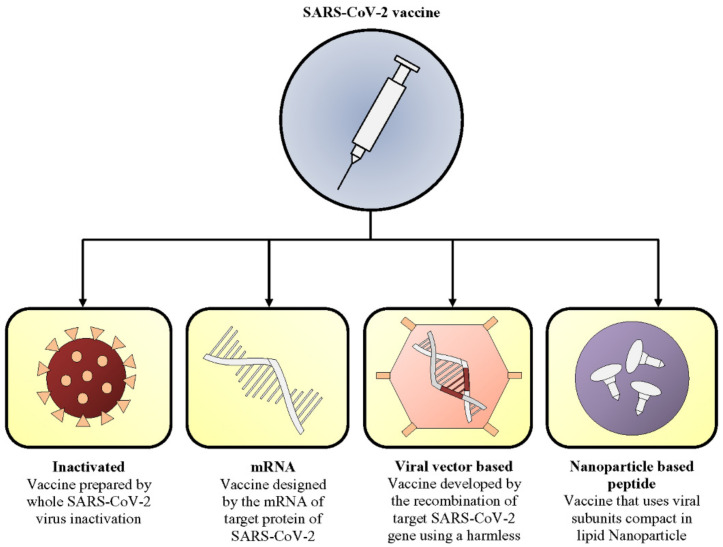
Types of COVID-19 vaccine developed based on different technologies.

**Figure 3 vaccines-09-01387-f003:**
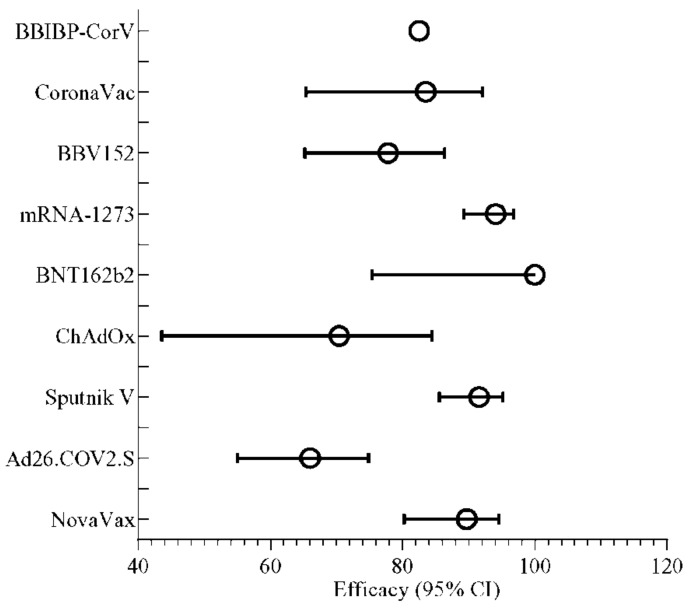
Efficacy of different SARS-CoV-2 vaccine candidates. Here, the small circles imply the reported efficacy after vaccination. All the data were extracted from the included articles, which were selected for this systematic review only. As all the vaccines did not have the same response levels, the 95% CIs were not evenly distributed. For BBIBP-CorV, we were not able to find the upper and lower limits of 95% CI; thus, it was not reported in the figure.

**Figure 4 vaccines-09-01387-f004:**
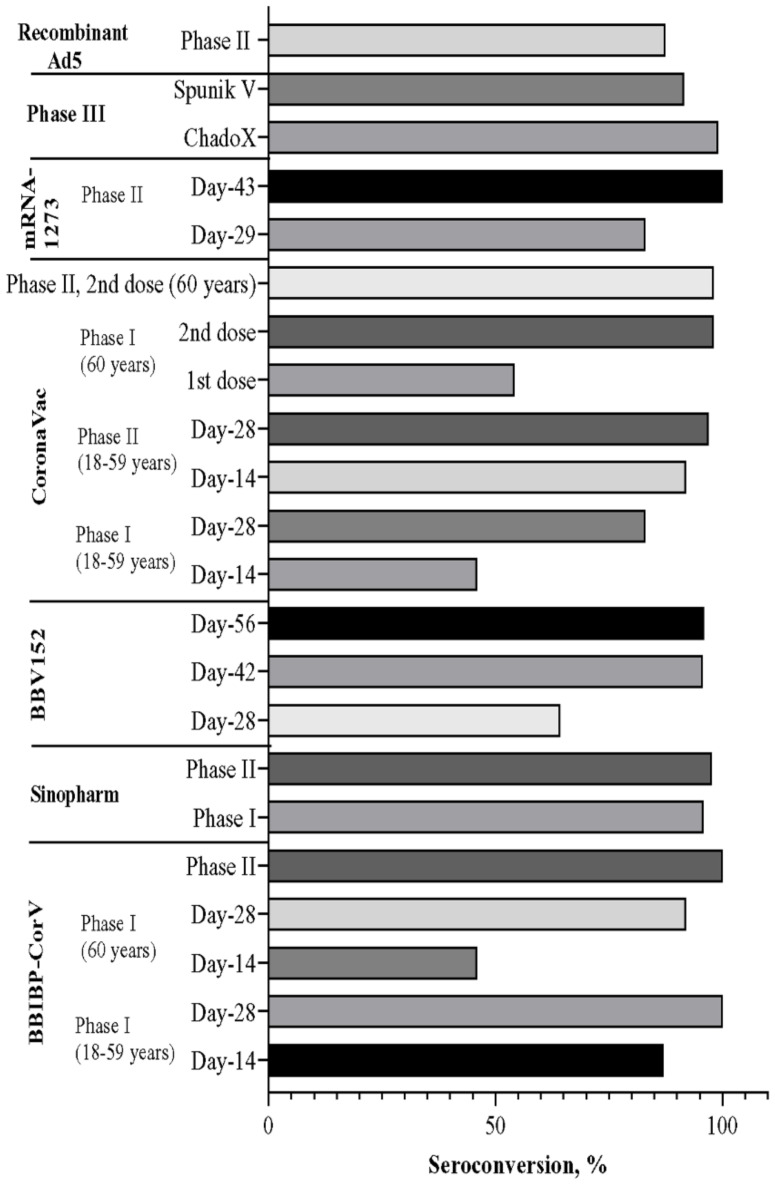
IgG seroconversion of several SARS-CoV-2 vaccines by trial Phase (i.e., Phase I/II/III), dose number (i.e., 1st or 2nd dose), or days after vaccination (i.e., day 14/28/29/42/56). Data were extracted from the included articles which were selected for this systematic review only.

**Figure 5 vaccines-09-01387-f005:**
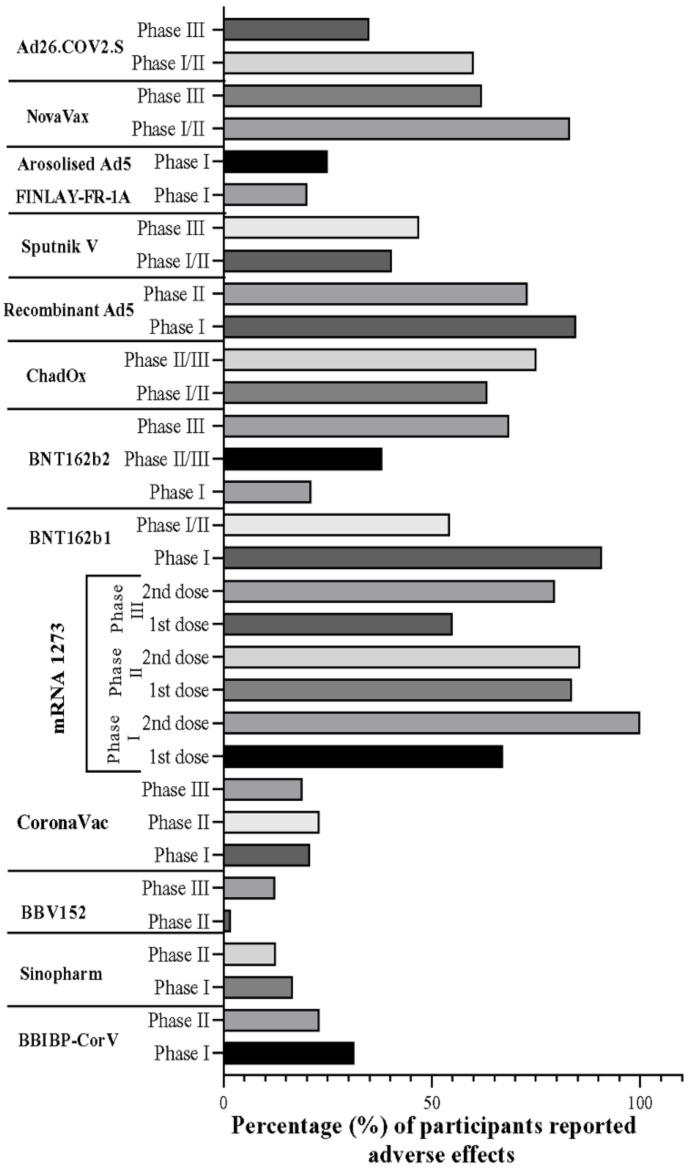
Adverse effects (AE) of several SARS-CoV-2 vaccines by trial phase (i.e., Phase I/II/III), dose number (i.e., 1st or 2nd dose). Data were extracted from the included articles which were selected for this systematic review only.

**Table 1 vaccines-09-01387-t001:** Quality assessment of the selected studies of clinical trial phase.

Study	1	2	3	4	5	6	7	8	9	10	11	12	13	14	Total Score (%)
Xia S., 2021	Y	Y	Y	Y	Y	Y	Y	Y	Y	Y	Y	NR	Y	Y	100
Zhang Y., 2021	Y	Y	Y	Y	Y	Y	Y	Y	Y	Y	Y	NR	Y	Y	100
Wu Z., 2021	Y	Y	Y	Y	Y	Y	Y	Y	Y	Y	Y	N	Y	Y	92.8
Tanriover M.D., 2021	Y	Y	Y	Y	Y	Y	Y	Y	Y	Y	Y	N	Y	Y	92.8
Ella R., Reddy, S., Jogdand, H, 2021	Y	Y	Y	Y	Y	Y	Y	Y	Y	Y	Y	Y	Y	Y	100
Ella R., R.; Reddy, S.; Blackwelder 2021	Y	Y	Y	Y	Y	Y	Y	Y	Y	Y	Y	Y	Y	Y	100
Xia S., 2020	Y	Y	Y	Y	Y	Y	Y	Y	Y	Y	Y	Y	Y	Y	100
Jackson L.A., 2020	NA	NA	N	N	N	Y	Y	Y	Y	Y	Y	NR	Y	NA	70
Chu L., 2021	Y	Y	Y	Y	Y	Y	Y	Y	Y	Y	Y	NR	Y	Y	100
Baden L.R., 2021	Y	Y	Y	Y	Y	Y	Y	Y	Y	Y	Y	NR	Y	Y	100
Mulligan M.J., 2020	Y	Y	Y	Y	Y	Y	Y	Y	Y	Y	Y	NR	Y	Y	100
Walsh E.E., 2020	Y	Y	Y	Y	Y	Y	Y	Y	Y	Y	Y	NR	Y	Y	100
Li J., 2021	Y	Y	Y	Y	Y	Y	Y	Y	Y	Y	Y	N	Y	Y	100
Polack F.P., 2020	Y	Y	Y	Y	Y	Y	Y	Y	Y	Y	Y	NR	Y	Y	100
Frenck Jr R.W., 2021	Y	Y	Y	Y	Y	Y	Y	Y	Y	Y	Y	NR	Y	Y	100
Chang-Monteagudo A., 2021	NA	NA	N	N	N	Y	Y	Y	Y	Y	Y	NR	Y	NA	70
Zhu F.C., Li, Y.H, 2020	N	NA	N	N	N	Y	Y	Y	Y	Y	Y	NR	Y	NA	63.6
Zhu F.C., Guan, X.H, 2020	Y	Y	Y	Y	Y	Y	Y	Y	Y	Y	Y	NR	Y	Y	100
Wu S., 2021	Y	Y	N	N	N	Y	Y	Y	Y	Y	Y	N	Y	Y	71.4
Folegatti P.M., 2020	Y	Y	Y	Y	Y	Y	Y	Y	Y	Y	Y	NR	NR	Y	100
Ramasamy M.N., 2020	Y	Y	Y	Y	Y	Y	Y	Y	Y	Y	Y	NR	NR	Y	100
Denis Y. Logunov, 2020	N	NR	NR	N	N	Y	Y	Y	Y	Y	Y	NR	NR	Y	70
Denis Y. Logunov, 2021	Y	Y	Y	Y	Y	Y	NR	NR	Y	Y	Y	Y	NR	Y	100
J. Sadoff, 2021	Y	Y	Y	Y	Y	Y	NR	NR	Y	Y	Y	NR	NR	Y	100
J. Sadoff, 2021	Y	Y	Y	Y	Y	Y	Y	Y	Y	Y	Y	Y	NR	Y	100
C. Keech, 2020	Y	Y	Y	NR	Y	Y	Y	Y	Y	Y	Y	Y	Y	Y	100
P.T. Heath, 2021	Y	Y	Y	NR	Y	Y	Y	Y	Y	Y	Y	Y	Y	Y	100
Ewer K.J.,2021	Y	Y	Y	NR	NR	Y	Y	Y	Y	Y	Y	NR	NR	Y	100
Barrett J.R.,2021	Y	Y	Y	Y	Y	Y	Y	Y	Y	Y	Y	NR	NR	Y	100

Here, Y = Yes, N = No, NR = Not reported, NA = Not applicable; 1. Was the study described as randomized, a randomized trial, a randomized clinical trial, or an RCT? 2. Was the method of randomization adequate (i.e., use of randomly generated assignment)? 3. Was the treatment allocation concealed (so that assignments could not be predicted)? 4. Were study participants and providers blinded to treatment group assignment? 5. Were the people assessing the outcomes blinded to the participants’ group assignments? 6. Were the groups similar at baseline on essential characteristics that could affect outcomes (e.g., demographics, risk factors, co-morbid conditions)? 7. Was the overall drop-out rate from the study at the endpoint 20% or lower than the number allocated to treatment? 8. Was the differential drop-out rate (between treatment groups) at endpoint 15 percentage points or lower? 9. Was there high adherence to the intervention protocols for each treatment group? 10. Were other interventions avoided or similar in the groups (e.g., identical background treatments)? 11. Were outcomes assessed using valid and reliable measures implemented consistently across all study participants? 12. Did the authors report that the sample size was sufficiently large to detect a difference in the main outcome between groups with at least 80% power? 13. Were outcomes reported or subgroups analyzed prespecified (i.e., identified before analyses were conducted)? 14. Were all randomized participants analyzed in the group to which they were originally assigned, i.e., did they use an intention-to-treat analysis?

**Table 2 vaccines-09-01387-t002:** The major characteristics of the reported COVID-19 vaccines in the trial phase.

Vaccine Type	Name	Manufacturer	Trials	Trial Model	Target	Efficacy	Advantages	Side Effects	Reference
**Inactivated**	BBIBP-CorV; Strain: HB02	Sinopharm	Randomized, double-blind, placebo controlled, phase 1/2 trial; 2 µg, 4 µg and 8µg dose of vaccine	Human (*n* = 192; phase 1) (*n* = 448; phase 2)	Whole virus	79–86% ^a^	Safe and well controlled; the humoral response was induced	Fever, pain, fatigue, nausea (Phase 1: 35.5% for age 18–59 years; 27% for 60+ years. Phase 2: 23% for age 18–59 years)	[120]
	CoronaVac; Strain: CN2	Sinovac life sciences Co., Ltd.	Randomized, double-blind, placebo controlled, phase 1 and 2 trial; 3 µg and 6µg dose of vaccine; Phase 3 trial in Turkey	Human; age: 18–59 years (*n*= 143; phase 1), (*n* = 600; phase 2), Age: 60 and older (*n* = 72; phase 1), (*n* = 350; phase 2); (age: 18–59 years; *n* = 10,281; phase 3)	Whole virus	83.5% (Phase 3)	The lower incident rate of side effects; Seroconversion rate over 90%; safe to administer even in older people	Pain at injection site, fever, fatigue (phase 1: 21%; Phase 2: 26% for 18–59 years) (phase 1: 20% for 60 years and older) (phase 3: 9.35%)	[122,123,124]
	BBV152; Strain: NIV-2020–770	Bharat Biotech	Phase 2, double-blind, randomized controlled trial, 3µg and 6 µg dose of vaccine; Phase 3, double-blind, randomized, controlled trial	Human (*n* =380; phase 2) (age 18–98 years; *n*= 25,798; phase 3)	Whole virus	63.6%, 77.8%, and 93.4% against asymptomatic, symptomatic and severe COVID-19 cases	The vaccine-induced both cellular and humoral immunity along with a long-lived memory response.	Pain at the injection site, fever, fatigue, headache, malaise, body ache, itching, weakness, redness at the injection site (Phase 2: 1.69%) (Phase 3: 12.4%)	[92,128]
	Inactivated whole virus COVID-19 vaccine; strain: WIV04	Sinopharm	Randomized, double-blind, placebo-controlled, phase 1 and 2 trial; Low (2.5 µg), medium (5µg) and high (10µg) for phase 1 trial and 5µg dose of vaccine for phase 2 trial	Human (*n* = 96; phase 1), (*n* =224; phase 2), ongoing study	Whole virus	NR	Immunogenic with a low occurrence of side effects	Pain in the injection site, fever, fatigue, nausea and vomiting (phase 1: 16.7%; Phase 2: 12.5%)	[127]
**mRNA**	mRNA-1273	Moderna	Phase 1 (dose escalation, open-label trial, 25 μg, 100 μg and 250 μg dose of vaccine) Phase 2,3 (randomized, observer-blind, placebo-controlled trial) phase 2: 50 or 100μg dose of vaccine Phase 3: 100μg dose of vaccine	Human (*n* = 45; Phase 1), (*n* = 600; Phase 2), (*n* = 30,420; Phase 3)	Spike glycoprotein (S-2P antigen)	94.1%	Immunogenicity is fast and powerful; antigen-specific T-follicular helper cells are induced by prolonged protein expression, and thus germinal center B cells are activated.	Fatigue, chills, fever, myalgia, and discomfort at the injection site were the cited adverse events; after the second dose, systemic adverse events were more frequent. (Phase 1: 67% after 1st dose and 100% after 2nd dose; Phase 2: 88% in younger and 81% in older; phase 3: 54.9% after 1st dose and 79.4% after 2nd dose)	[158,159,160]
	BNT162b1	BioNTech and Pfizer	Phase 1, Phase 1/2 (placebo-controlled, observer-blinded, dose-escalation trial) Phase 1: 10 μg, 20 μg, 30 μg, and 100 μg dose of vaccine (In U.S.) Phase I/II: 10μg, 30μg or 100μg dose of vaccine Phase 1 (randomized, placebo-controlled, double-blind trial): 10 μg or 30 μg dose of vaccine (In Chinese participants)	Human (*n* = 195; Phase 1 in U.S.), (*n* = 45; Phase I/II), (*n* = 144; Phase 1 in Chinese participants)	RBD of the spike protein	NR	Immune-stimulatory; can be delivered into cells more effectively; elicit both humoral and cell-mediated antiviral mechanisms	Pain at the injection site, fatigue, headache, chills, muscle pain, joint pain; in older adults, systemic reactogenicity profile is severe (Phase 1/2: 54.2%); (Phase 1 study of Chinese participants: 88% (10 μg) and 100% (30 μg) in younger participants; 83% (10 μg) and 92% (30 μg) in older participants	[130,161,163]
	BNT162b2	BioNTech and Pfizer	Phase 1 (placebo-controlled, observer-blinded, dose-escalation trial, 10 μg, 20 μg, 30 μg, and 100 μg dose of vaccine); Phase 2/3 (ongoing multinational, placebo-controlled, observer-blinded, pivotal efficacy the trial, 30 μg dose of vaccine); Phase 3 (ongoing multinational, placebo-controlled, observer-blinded trial, 30 μg dose of vaccine)	Human (*n* =195; Phase 1), (*n* = 43,548; Phase 2/3) (*n* = 2260; Phase 3)	Full-length spike	95% (16 years of age or older) 100% (12 to 15 years of age)	In older adults, systemic reactogenicity profile is mild mainly; reactogenicity and immunogenicity profile are in a good balance; Highly effective against COVID-19 in adolescents	Pain at the injection site, fatigue, chills, muscle pain, joint pain, headache, fever, redness or swelling (Phase 1: 17% in 65–85 years. 25% in 18–55 years; Phase 2/3: 42.33% in younger and 33.67% in older; Phase 3: 68.5%)	[94,161,164]
**Recombinant**	FINLAY-FR-1A	Finlay Vaccine Institute in Havana, Cuba	Phase 1 (open, adaptive, and monocentric clinical trial, 50μg dose of vaccine) Ongoing	Human (*n* = 30)	RBD	NR	Excellent safety profile and single-dose increases neutralization responses in COVID-19 convalescents	Pain at the injection site, warmth, redness, swelling, malaise, rash, fever, high blood pressure. (Phase 1: 20%)	[194]
	Ad5 vectored COVID-19 vaccine	Beijing Institute of Biotechnology and CanSino Biologics	Phase 1 (dose escalation, single-center open-label, non-randomized trial, dose: 5 × 10^10^, 1 × 10¹¹, and 1·5 × 10^¹¹^ viral particles); Phase 2 (randomized, double-blind, placebo-controlled trial, dose: 1 × 10^¹¹^ viral particles per mL or 5 × 10^¹⁰^ viral particles per mL)	Human (*n* = 108; Phase1) (*n* = 508; Phase 2)	Spike glycoprotein	NR	Well tolerable and immunogenic; Safe; significant immune responses are induced after a single vaccination.	Systematic adverse reactions: fever, fatigue, headache, and muscle pain (Phase 1: 84.5%; phase 2: 73% solicited adverse event and 5% severe adverse reaction)	[178,179]
	Aerosolised Ad5-nCoV	Institute of Biotechnology, Academy of Military Medical Sciences, PLA of China	Phase 1 (randomized, single-center, open-label, trial, dose: 2 × 10^10^, 1 × 10^10^, 5 × 10^10^, 10 × 10^10^ viral particles)	Human (*n* = 130)	Spike glycoprotein	NR	Well tolerable, Painless, Simple, Strong IgG and neutralizing antibody responses.	Fever, fatigue, headache (25% adverse events in aerosol vaccine group)	[180]
	ChAdOx1 nCoV-19 OR AZD1222	AstraZeneca	Phase 1/2 single-blind, randomized controlled trial receive ChAdOx1nCoV-19 or MenACWY at a dose of 5 × 10¹⁰ viral particles; Phase 2/3 trial; LD cohort -receive; 2·2 × 10¹⁰ virus particles; SD cohort receive 3·5–6·5 × 10¹⁰ virus particles of ChAdOx1 nCoV-19)	Healthy human model (Phase 1/2, *n* = 1077; Phase 2/3, *n* = 560	Whole Spike protein	Overall efficacy—70.4%	Single-dose of ChAdOx1 nCoV-19 elicits increased spike-specific antibody; remain asymptomatic to develop a robust memory T-cell response; safe, tolerated, and immunogenic; elicit both humoral and cellular responses; no adverse events even after the booster dose	Local and systemic reactions including; injection site pain, feverish, chills, muscle ache, headache, and malaise. Phase 1/2 trial—63.259%; Phase 2/3 trial- 75%	[93,181,182,183]
	Sputnik V, (Gam-COVID-Vac)	Developed by The Gamaleya National Center of Epidemiology and Microbiology	Phase 1/2 studies at two hospitals in Russian two studies (38 in each study). In each study, nine volunteers received rAd26-S in phase 1, nine received rAd5-S in phase 1, and 20 received rAd26-S and rAd5-S in phase 2. Phase 3 trial held at 25 hospitals	Human model (phase 1/2, *n* = 76 Phase 3, *n* = 21,977 (Vaccine group *n*-16,501; Placebo group *n*-5476)	Spike protein Vector	91.6% after 2 doses, 79.4% after 1 dose	Safe; well tolerated; induced strong humoral and cellular immune responses in 100% of healthy participants; no serious adverse events	The most common systemic and local reactions were pain at the injection site, hyperthermia, headache, asthenia, and muscle and joint pain; Phase 1/2 trial—40.4%; Phase 3 trial, 47%	[187,193]
	Ad26.COV2.S	Manufactured by Janssen Pharmaceuticals companies acquired by Johnson & Johnson	Multicenter, placebo-controlled, Phase 1–2a trial to evaluate the safety and immunogenicity profiles of Ad26.COV2.S; Randomized, double-blind, placebo-controlled, Phase 3 trial to determine the effectiveness of the vaccine.	Human model (Phase 1/2a trial, *N*, 805 participants Phase 3 trial, *n*, 43,783)	A recombinant, replication-deficient adenovirus serotype 26 (Ad26) vector encoding a stabilized SARS-CoV-2 spike (S) protein	66%	Neutralizing antibody response 100% by day 57; Spike-binding antibody and neutralizing antibody response were similar; CD4+ T-cell response- 76–83% (18–55 years age group) 60–66% (65 years or older)	Cohort 1(18–55 years group): local adverse event 71% and systemic adverse event-74.5%; Cohort 3 (65 years or older): local adverse event 41.5% and systemic adverse event 50.5%	[196,198]
**Nanoparticles**	NVX-CoV2373	Novavax	A randomized, placebo-controlled, Phase 1–2 trial to evaluate the safety and immunogenicity of the rSARS-CoV-2 vaccine (in 5 μg and 25 μg doses, with or without Matrix-M1 adjuvant Phase 3, randomized, observer-blinded, placebo-controlled trial conducted at 33 sites in the UK	Human model, (Phase 1/2a, *n*-131; Phase 3, *n*-15,187)	Nanoparticle vaccine composed of trimeric full-length SARS-CoV-2 spike glycoproteins and Matrix-M1, a saponin-based adjuvant Vector type- Baculovirus	89.7%	Proper vaccine-induced immunogenicity; Stimulates both high neutralizing antibody responses and T cells; storable at 4 degrees C for a long time and easily transportable; capable of restraining new variants in UK and South Africa	Phase 1/2 trial-localized symptoms: (86.4%) erythema or redness, induration or swelling, pain, tenderness; systemic symptoms: (79.4%) arthralgia, fatigue, fever, headache, myalgia, nausea, malaise; Phase 3 trial—Systemic adverse event; after 1st dose 21.76%. After 2nd dose—40.2%	[208,209]

^a^ [270,271], NR = Not reported.

## Data Availability

Not applicable.

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
