# Peer review of "A Systematic Review on COVID-19 Vaccine Strategies, Their Effectiveness, and Issues"

_vaccines, 2021, doi:10.3390/vaccines9121387_

Round 1

Reviewer 1 Report

This review of clinical assesments of SARS-CoV2 vaccines appears comprehensive, but I found it difficult to read. Descriptions of the underlying vaccine technologies, in particular, lacked clear organising structure and suffered from extensive typological and grammatical errors. 

Table 1 lists 27 studies, but the text implies that 30 should be present

Line 240 "the safety profile ... was noticeably lower". It is not clear what this means. Is a 'lower safety profile' better or worse? Differences in safety profile should be described in more detail. In particular, the means used to map side-effects to a 0-100 scale for Fig 5 must be outlined. Does this figure present side-effect frequency or severity? The complete lack of any discussion of vaccine-associated thrombosis with thrombocytopenia is an inexplicable omission.

The presentation of Figs 3, 4 & 5 is problematic in several respects. Fig 3 shows CoronaVac to have higher overall efficacy than BBIBP-CorV or BBV152, yet the text claims the opposite (Line 244). What is the significance of the color gradients here? A more conventional presentation clearly showing estimated quantities with confidence intervals would be much more useful. Dark text on dark background is hard to read. 

Author Response

Manuscript ID: vaccines-1421083

Title: A Systematic Review on COVID-19 Vaccine Strategies, their effectiveness, and Issues

Reviewer 1:

Open Review

(x) I would not like to sign my review report

( ) I would like to sign my review report

English language and style

(x) Extensive editing of English language and style required

( ) Moderate English changes required

( ) English language and style are fine/minor spell check required

( ) I don't feel qualified to judge about the English language and style

Yes      Can be improved        Must be improved      Not applicable

Does the introduction provide sufficient background and include all relevant references?

( )        (x)       ( )        ( )

Is the research design appropriate?

( )        ( )        (x)       ( )

Are the methods adequately described?

( )        ( )        (x)       ( )

Are the results clearly presented?

( )        ( )        (x)       ( )

Are the conclusions supported by the results?

( )        (x)       ( )        ( )

Comments and Suggestions for Authors

This review of clinical assesments of SARS-CoV-2 vaccines appears comprehensive, but I found it difficult to read. Descriptions of the underlying vaccine technologies, in particular, lacked clear organising structure and suffered from extensive typological and grammatical errors.

Answer:

Thank you for the comments. We understand the reviewer's difficulty in following the manuscript as we tried to perform the study thoroughly. We have restructured the manuscript and vigorously edited the manuscript, and corrected the typological and grammatical errors. We again like to thank the reviewer for his suggestions to improve the quality of the manuscript.

Table 1 lists 27 studies, but the text implies that 30 should be present.

Answer:

Thank you so much for this critical observation. Two errors occurred during the writing. One is that the number of included clinical trials would be 29, rather than 30, and the number of pre-/non clinical trials or other related research articles would be 30, rather than 29. The corrections have been made in the methodology section (Line-142-143) and Figure 1 as well. Again, two of the missing articles of the clinical trial phase have been added in Table-1 of the quality assessment (in last two rows), which we, unfortunately, missed while separating the files. Again thanks a lot for correcting these unfortunate errors.  

Line 240 "the safety profile ... was noticeably lower". It is not clear what this means. Is a 'lower safety profile' better or worse? Differences in safety profile should be described in more detail. In particular, the means used to map side-effects to a 0-100 scale for Fig 5 must be outlined. Does this figure present side-effect frequency or severity? The complete lack of any discussion of vaccine-associated thrombosis with thrombocytopenia is an inexplicable omission.

Answer:

Thank you so much for your valuable suggestion. A lower safety profile indicates that the vaccine-induced adverse reaction is lower and better for a vaccine candidate. We changed it in the manuscript (Line-260-262).

The scaling of figure 5 indicates the percentage. To avoid any confusion, we have revised figure 5, horizontally showing the percentage of side effects reported in the literature on each vaccine. This is to be mentioned that these side effects are mainly minor side effects.

We agree with the reviewer that severe side effects like thrombosis with thrombocytopenia and others were not mentioned in the initial manuscript. Out initial thought was, mentioning the severe side effects would create vaccine hesitancy among the readers. But we agree with the reviewer that these side effects should be mentioned for proper comparison among vaccines, and we have edited the manuscript accordingly (Line-80).

The presentation of Figs 3, 4 & 5 is problematic in several respects. Fig 3 shows CoronaVac to have higher overall efficacy than BBIBP-CorV or BBV152, yet the text claims the opposite (Line 244). What is the significance of the color gradients here? A more conventional presentation clearly showing estimated quantities with confidence intervals would be much more useful. Dark text on dark background is hard to read.

Answer:

Thank you very much for performing a thorough review of our manuscript. We made a mistake while updating the efficacy rate of the vaccines. The phase 3 trial of CoronaVac says that its efficacy rate is “83.5% after the second dose”, updated in the manuscript (Line-265-267).

We understand the reviewer's difficulty in understanding figures 3, 4, and 5. As we planned to perform systematic analysis, not a meta-analysis, we didn't calculate the estimated quantities with a confidence interval for efficacies, side-effects, and seroconversion. But, we agree with the reviewer for better comparison among vaccines, this type of analysis is important, and we have edited figure 3 with quantities with confidence intervals reported in the selected literature. For figures 4 and 5, no such data were found in the selected literature. Thus we revised the figures according to the reviewer's suggestions and showed the data on a percentage scale of 0-100 without gray-scale. We again like to thank the reviewer for the critical assessment of our manuscript.

Submission Date

29 September 2021

Date of this review

15 Oct 2021 06:42:24

Reviewer 2:

Open Review

( ) I would not like to sign my review report

(x) I would like to sign my review report

English language and style

( ) Extensive editing of English language and style required

(x) Moderate English changes required

( ) English language and style are fine/minor spell check required

( ) I don't feel qualified to judge about the English language and style

Yes      Can be improved        Must be improved      Not applicable

Does the introduction provide sufficient background and include all relevant references?

( )        (x)       ( )        ( )

Is the research design appropriate?

( )        (x)       ( )        ( )

Are the methods adequately described?

( )        ( )        (x)       ( )

Are the results clearly presented?

( )        (x)       ( )        ( )

Are the conclusions supported by the results?

( )        (x)       ( )        ( )

Comments and Suggestions for Authors

Khandker and coworkers have undertaken a systematic literature review of the different vaccines and vaccine strategies against COVID-19. This work is of great importance, as it is not only the vaccines themselves, but how they are administered, combined and spaced will determine our success in managing a world-wide SARS-CoV-2 pandemic. The authors didn't always use the best methodology, but they did a very thorough examination of the published material and presented it logically and systematically. I have some suggestions that should help the authors improve their manuscript.

Major points

The authors refer to COVID-19 as a respiratory infection. This is, of course, mostly true. However, emerging evidence suggests that although the primary infection is in the upper respiratory tract, the widespread expression of the ACE2 receptor across multiple tissues may be responsible for most long-term effects. This should be discussed.

Answer:

Thank you for your valuable suggestion. SARS-CoV-2, like its predecessor SARS-CoV, has the pathophysiology of multi-organ damage. We explained the presence of the ACE2 receptor in cells of other organs and briefly discussed the pathophysiology of SARS-CoV-2 (Line-74-88).

I also take issue with the statement that SARS-CoV-2 does not stimulate the innate immune response. I seriously doubt this to be true. Although the authors do provide good references for this, my interpretation of these results is that the virus is suppressing the innate immune response (particularly interferon inducing proteins), rather than simply escaping the attention of the mechanisms inside the cell. Indeed, it is this "race" between the ability of the virus to suppress the intracellular signalling and the efficiency of the signalling process that determines whether the immune system is activated early enough to result in asymptomatic disease vs symptomatic disease. Indeed, this would explain why the elderly and people with comorbidities are more likely to become very ill, even though the disease is a result of immune over-reaction, which one would not expect in the elderly and sick.

Answer: We would like to thank the reviewer for these valuable suggestions. We also agree with the reviewer that suppression of signaling pathways leading to interferon release in mild covid-19 cases is responsible for asymptomatic cases, whereas shifting of neutrophilia over lymphopenia is the leading cause of cytokine storms and severe covid. We have discussed reviewers ideas in the manuscript (Line-81-88)

Quite a few of the basic systematic review methods have not been reported. I believe that they have been done, but they must be reported properly. The PRISMA checklist is a great tool to ensure everything is reported. They can just leave out the meta-analysis parts. Things missing include (but may not be limited to):

Actual search strategies (not just the words searched)

Whether the review was pre-registered (and where, and the number)

Inclusion and exclusion criteria

The method and technology used for including and excluding studies, how many authors did it, and how they resolved discrepancies

For any data that were extracted, whether this was checked by a second author and how discrepancies were resolved

The technology used for collating the data that was extracted

The technology used for citation management

Answer: Thanks a lot for the valuable queries and suggestions. We did modify the methodology according to your comments including.

The actual search strategy is shared in the supplementary file.

We tried to register this study in PROSPERO, but as we had started data extraction from the selected articles before registration, we finally could not register it.

The inclusion and exclusion criteria are described in the methodology section (Line-127-133) as well as in revised Figure 1.

The data extraction method and assessment steps are briefly described in the method section (Line-134-142).

We have also mentioned the technology used for collating the data and citation management in the method section (Line-143-144).

Table 2 was extremely good and will be a very useful reference for everyone involved with SARS-CoV-2 vaccines. The table title should be changed to include SARS-CoV-2, as the authors also talk about other coronaviruses in their manuscript.

Answer:

Thanks a lot for your suggestion. "SARS-CoV-2" has been included in the title of table-2. 

Figure 2 is rather poorly done. Some money should be spent to get a proper scientific illustrator to make this look good.        

Answer:

We would like to thank the reviewer for the suggestion. Unfortunately, we couldn't take professional service, as this work was done without any funding. Thus authors couldn't afford to take external help. To the best of our ability, we have modified figure 2 to make it a proper scientific illustration.

Figure 3 is confusing. Firstly, the use of colour is not clear and should be explained. Secondly, the terms "overall" and "Phase I" are not explained. Thirdly, the fading at the end of the efficacy part isn't explained. Is this just to look good? Does it represent the 95% confidence intervals? The ranges between studies? Interquartile ranges? Something else?

Answer:

Thank you so much for your kind concern. We agree with the reviewer that figure 3 was confusing and have revised the figure. Figure 3 has been modified accordingly, and the term "overall" has been removed. As the reviewer knows, we didn't perform a meta-analysis. We couldn't estimate the quantities with a confidence interval. In the revised figure 3, we have shown the quantities with confidence intervals mentioned in the selected articles.

The same goes for figures 4 and 5. Now the efficacy bars are orange and watermelons colours, respectively. The authors should find a colour scheme and stick to it and explain it to the audience. Their figure legends should be so informative that a person should be able to understand it without reading any of the text.

Answer:

We would like to thank the reviewer for the suggestions. The figures have been updated accordingly (revised figures 4 and 5). We believe now the figures are more informative and readable. Revision of these figures has significantly improved the manuscript's quality, and we would like to thank the reviewer again for pointing it out.

I would like to see a figure comparing the efficacy of each vaccine for each of the variants. It would give us an easy way of seeing whether certain vaccines, or vaccine types, provide better coverage across variants, or whether they all drop in efficacy. Just don't use crazy colours!!

Answer:

Thanks a lot for the comment. This really would be an excellent addition and would enhance the value of this manuscript. However, unfortunately, not enough data were found regarding the variant-specific vaccine efficacy from the selected articles. Therefore we could not provide any figure regarding this issue. But we have mentioned some of the efficiencies in the manuscript (Line-787-824). But, we agree with the reviewer, variant-specific protection by different vaccines could significantly attract more readers.

In the section on gender-based toxic effects, it is VERY important to state in lines 813-820 that these were unfounded speculations. As the authors stated, the vaccines are absolutely safe in pregnancy and indeed provide protection to the infant. Vaccines against COVID-19 should be encouraged for pregnant women, because they also do badly if they get infected during pregnancy. This should be mentioned.

Answer:

Thank you so much for your kind suggestions. The section has been modified according to your suggestion (Line-853-863).

I think the single example of the 56 year-old woman should be removed. If you vaccinate billions of people, every now and then something odd is going to happen. The authors should be focussing on the most common serious and non-serious adverse events. For example, a large number of women report alterations in their menstrual cycle after receiving the mRNA vaccines, but it is entirely unclear if this is actually related to the vaccine or not.

Answer:

Thank you very much for your valuable suggestion. The single example has been removed and the concern for adverse events has been mentioned as per your recommendation (Lines-864-893)

The authors should look closely for any articles on vaccine mixing. I believe that a prime-boost strategy could have an even better outcome than simply getting two or more doses of the same vaccine. However, I am not aware of any published trials. If they can't find any, they should at least include a paragraph on the concept of prime-boost and why it would be important to do trials in this area.

Answer:

Thanks a lot for your kind suggestion. We have added a mix and match section in the manuscript according to your valuable suggestion (Lines-917-931).

Minor points

The English in the manuscript is good but would benefit from a specialist scientific proof-reader.

Answer:

We like to thank the reviewer for the suggestion, and we have extensively edited the manuscript accordingly.

Line 86: lockdowns and social distancing measures are not "treatments" per se, they are public health measures.

Answer: Thank you so much for this suggestion. The correction has been made accordingly (Lines-94-95).

The database search for articles was not really thorough, in the sense that the authors didn't include clinical trial databases such as the WHO clinical trials registry and clinicaltrials.gov. In addition, the Cochrane Library and EBSCO etc were not searched. If this had been a meta-analysis, I would have insisted they redo the search, but considering that a) the COVID vaccine trials will be pretty obvious and easy to find, and b) this is a literature review, I won't make the authors redo their search. However, they should, in the future take a very high quality meta-analysis and follow the same methodology.

Answer:

We would like to thank you for your valuable advice and acknowledge our limitations. We would try to keep this suggestion in mind while conducting related systematic reviews and meta-analyses in the future.

/

Submission Date

29 September 2021

Date of this review

17 Oct 2021 04:19:00

Reviewer 2 Report

Khandker and coworkers have undertaken a systematic literature review of the different vaccines and vaccine strategies against COVID-19. This work is of great importance, as it is not only the vaccines themselves, but how they are administered, combined and spaced will determine our success in managing a world-wide SARS-CoV-2 pandemic. The authors didn’t always use the best methodology, but they did a very thorough examination of the published material and presented it logically and systematically. I have some suggestions that should help the authors improve their manuscript.

Major points

The authors refer to COVID-19 as a respiratory infection. This is, of course, mostly true. However, emerging evidence suggests that although the primary infection is in the upper respiratory tract, the widespread expression of the ACE2 receptor across multiple tissues may be responsible for most long-term effects. This should be discussed.

I also take issue with the statement that SARS-CoV-2 does not stimulate the innate immune response. I seriously doubt this to be true. Although the authors do provide good references for this, my interpretation of these results is that the virus is suppressing the innate immune response (particularly interferon inducing proteins), rather than simply escaping the attention of the mechanisms inside the cell. Indeed, it is this “race” between the ability of the virus to suppress the intracellular signalling and the efficiency of the signalling process that determines whether the immune system is activated early enough to result in asymptomatic disease vs symptomatic disease. Indeed, this would explain why the elderly and people with comorbidities are more likely to become very ill, even though the disease is a result of immune over-reaction, which one would not expect in the elderly and sick.

Quite a few of the basic systematic review methods have not been reported. I believe that they have been done, but they must be reported properly. The PRISMA checklist is a great tool to ensure everything is reported. They can just leave out the meta-analysis parts. Things missing include (but may not be limited to):

  • Actual search strategies (not just the words searched)
  • Whether the review was pre-registered (and where, and the number)
  • Inclusion and exclusion criteria
  • The method and technology used for including and excluding studies, how many authors did it, and how they resolved discrepancies
  • For any data that were extracted, whether this was checked by a second author and how discrepancies were resolved
  • The technology used for collating the data that was extracted
  • The technology used for citation management

Table 2 was extremely good and will be a very useful reference for everyone involved with SARS-CoV-2 vaccines. The table title should be changed to include SARS-CoV-2, as the authors also talk about other coronaviruses in their manuscript.

Figure 2 is rather poorly done. Some money should be spent to get a proper scientific illustrator to make this look good.         

Figure 3 is confusing. Firstly, the use of colour is not clear and should be explained. Secondly, the terms “overall” and “Phase I” are not explained. Thirdly, the fading at the end of the efficacy part isn’t explained. Is this just to look good? Does it represent the 95% confidence intervals? The ranges between studies? Interquartile ranges? Something else?

The same goes for figures 4 and 5. Now the efficacy bars are orange and watermelons colours, respectively. The authors should find a colour scheme and stick to it and explain it to the audience. Their figure legends should be so informative that a person should be able to understand it without reading any of the text.

I would like to see a figure comparing the efficacy of each vaccine for each of the variants. It would give us an easy way of seeing whether certain vaccines, or vaccine types, provide better coverage across variants, or whether they all drop in efficacy. Just don’t use crazy colours!!

In the section on gender-based toxic effects, it is VERY important to state in lines 813-820 that these were unfounded speculations. As the authors stated, the vaccines are absolutely safe in pregnancy and indeed provide protection to the infant. Vaccines against COVID-19 should be encouraged for pregnant women, because they also do badly if they get infected during pregnancy. This should be mentioned.

I think the single example of the 56 year-old woman should be removed. If you vaccinate billions of people, every now and then something odd is going to happen. The authors should be focussing on the most common serious and non-serious adverse events. For example, a large number of women report alterations in their menstrual cycle after receiving the mRNA vaccines, but it is entirely unclear if this is actually related to the vaccine or not.

The authors should look closely for any articles on vaccine mixing. I believe that a prime-boost strategy could have an even better outcome than simply getting two or more doses of the same vaccine. However, I am not aware of any published trials. If they can’t find any, they should at least include a paragraph on the concept of prime-boost and why it would be important to do trials in this area.

Minor points

The English in the manuscript is good but would benefit from a specialist scientific proof-reader.

Line 86: lockdowns and social distancing measures are not “treatments” per se, they are public health measures.

The database search for articles was not really thorough, in the sense that the authors didn’t include clinical trial databases such as the WHO clinical trials registry and clinicaltrials.gov. In addition, the Cochrane Library and EBSCO etc were not searched. If this had been a meta-analysis, I would have insisted they redo the search, but considering that a) the COVID vaccine trials will be pretty obvious and easy to find, and b) this is a literature review, I won’t make the authors redo their search. However, they should, in the future take a very high quality meta-analysis and follow the same methodology.

Author Response

(The authors gave the same response as above.)

Round 2

Reviewer 1 Report

The revisions made to the manuscript have markedly improved the overall presentation. However side effects and AEs remain poorly treated and must be corrected prior to publication.

I remain unclear about the meaning of the side-effects scale, Fig 5. Is this the %age of trial participants who reported any form of AE? Or does the scale have some other meaning?

Lines 865-866: this sentence seems to imply that persons older than 65 y did not experience adverse events. I assume this is not what the authors mean to say, and that a word or phrase is missing from this sentence.

Lines 869-870: these percentages imply a very much greater level of SAE than are reasonably associated with these vaccines. These numbers simply must be better contextuallised. What is the source of these reports? How many reports, and from how many total vaccinees? Given that Fig 5 could be read to state that these vaccines have AE rates in excess of 50%, the suggestion that 2% of these AEs are fatal overstates the relevant rate by many orders of magnitude.

Lines 877-878: How was this selection made? On what basis are two-thirds of reported AEs excluded from consideration?

Line 879: states that 28799 is 27.7% of 8007. What do the authors really mean here?

As noted in my original review, any discussion of the safety profile of COVID vaccines must address the thrombosis with throbocytopaenia syndrome associated very rarely with ChAdOx1 nCoV-19 immunisation. This is widely perceived (correctly or otherwise) to be the most important SAE associated with COVID vaccination. To ignore it in this context is simply not credible.

Author Response

First Reviewer

The revisions made to the manuscript have markedly improved the overall presentation. However side effects and AEs remain poorly treated and must be corrected prior to publication.

I remain unclear about the meaning of the side-effects scale, Fig 5. Is this the %age of trial participants who reported any form of AE? Or does the scale have some other meaning?

Ans: We thank the reviewer for the observation. In Fig 5., the "side effects %" actually represents the "percentage participants reporting AEs". We have made the necessary correction within the manuscript for a better understanding of the figure. We have also made the required correction to the legend within Fig 5.

Lines 865-866: this sentence seems to imply that persons older than 65 y did not experience adverse events. I assume this is not what the authors mean to say, and that a word or phrase is missing from this sentence.

Ans: We thank the reviewer for their comment. Based on the reference cited, we have rephrased the sentence to reflect the actual finding that we wished to convey. (Lines 849-854)

Lines 869-870: these percentages imply a very much greater level of SAE than are reasonably associated with these vaccines. These numbers simply must be better contextuallised. What is the source of these reports? How many reports, and from how many total vaccinees? Given that Fig 5 could be read to state that these vaccines have AE rates in excess of 50%, the suggestion that 2% of these AEs are fatal overstates the relevant rate by many orders of magnitude.

Ans: We thank the reviewer for the comment. Based on the reference cited, we have rephrased the sentence to reflect the actual findings that we wished to convey. (Line 856-859)

Lines 877-878: How was this selection made? On what basis are two-thirds of reported AEs excluded from consideration?

Ans: We thank the reviewer for their observation. The study from which we cited the numbers collected 32054 subjects from the Vigibase database. These subjects reported a total of 103954 AEs, at a rate of 3.24 AEs per subject. There was no exclusion in this particular case. The database itself has 32054 subjects as of 24th January 2021, as stated by the authors of the article. We have rephrased our sentences to reflect the actual findings better. (Lines 862-865)

Line 879: states that 28799 is 27.7% of 8007. What do the authors really mean here?

Ans: We have rephrased the sentence to convey our message better. 28799 is 27.7% of 103954 AEs that were reported. These 28799 AEs were characterized as SAEs, which were all among the 8007 subjects out of 32054 subjects that were studied. (Line 865-867)

As noted in my original review, any discussion of the safety profile of COVID vaccines must address the thrombosis with throbocytopaenia syndrome associated very rarely with ChAdOx1 nCoV-19 immunisation. This is widely perceived (correctly or otherwise) to be the most important SAE associated with COVID vaccination. To ignore it in this context is simply not credible.

Ans: We thank the reviewer for the suggestion. We have included a paragraph within the manuscript to shed a better light on thrombocytopenia syndrome. (Line 882-895)

Reviewer 2 Report

The authors have done a great job of improving their manuscript. Figure two looks great now, and the other figures are also much improved. I only have a few tiny changes now.

1) Figure 3: The data look to me like they are medians, not means, because the 95% CIs are not evenly distributed around the circles. Please double check. Also the legend text should be fixed. Something like: "Median (95% CI) efficacy of several SARS-CoV-2 vaccines. Circles indicate the median efficacy in prevention of severe disease or hospitalisation."

2) Figure 4: The figure legend is not good. It should be something like "IgG seroconversion of several SARS-CoV-2 vaccines by trial phase, dose number, or days after vaccination."

3) Figure 5: fix the legend to be like the ones I suggested above.

The "gender-based adverse effects" section needs some work still. The current wording suggests that the mRNA vaccines DO HAVE a toxic effect, whereas you are actually saying that some people suggested this, but it if unfounded. You should write something like: "Some concerns have been raised about the safety of COVID-19 vaccine. For example, it was suggested that the vaccines could have an impact of pregnancy and/or damage fertility. In particular, mRNA vaccines were claimed to be cross-reactive with the human placental protein syncytin, potentially causing placental damage."    Then you can continue with your paragraph.

The new manuscript needs to be proofread for English, as the English is good but sometimes a bit confusing. 

I absolutely applaud the authors for their enormous effort to make their manuscript significantly better. They did not try to do just the bare minimum - instead they did an excellent job of following the suggestions of the reviewers. Well done~!

Author Response

Second Reviewer

The authors have done a great job of improving their manuscript. Figure two looks great now, and the other figures are also much improved. I only have a few tiny changes now.

1) Figure 3: The data look to me like they are medians, not means, because the 95% CIs are not evenly distributed around the circles. Please double check. Also the legend text should be fixed. Something like: "Median (95% CI) efficacy of several SARS-CoV-2 vaccines. Circles indicate the median efficacy in prevention of severe disease or hospitalisation."

Ans: Thanks to the reviewer for this comment. The figure reported the efficacy with a 95% confidence interval (CI) of each vaccine from the published literature. The circle represents the estimated efficacy after vaccination, i.e., how many participants responded with seroconversion against SARS-CoV-2 after being vaccinated. The majority of the vaccine showed efficacy within 90%-100%, while only a few were below 80%. Since the seroconversion within each vaccine group was not always homogenous, the vaccine's efficacy is represented as 95% CI, which signifies the unevenness of a sample response for a particular vaccine group. For example, in the case of BBIBP-CorV, we could not find the upper and lower limits of 95% CI within our literature search; thus, it was not reported in the figure.

2) Figure 4: The figure legend is not good. It should be something like "IgG seroconversion of several SARS-CoV-2 vaccines by trial phase, dose number, or days after vaccination."

Ans:  We thank the reviewer for the suggestion. We have incorporated the changes to the legend within the manuscript.

3) Figure 5: fix the legend to be like the ones I suggested above.

Ans: We thank the reviewer for the suggestion. As previously suggested, we have incorporated the changes to the legend within the manuscript.

The "gender-based adverse effects" section needs some work still. The current wording suggests that the mRNA vaccines DO HAVE a toxic effect, whereas you are actually saying that some people suggested this, but it if unfounded. You should write something like: "Some concerns have been raised about the safety of COVID-19 vaccine. For example, it was suggested that the vaccines could have an impact of pregnancy and/or damage fertility. In particular, mRNA vaccines were claimed to be cross-reactive with the human placental protein syncytin, potentially causing placental damage."    Then you can continue with your paragraph.

Ans: We thank the reviewer for their critical observation and their suggestion. We have included the suggested sentence within the manuscript. (Line 833-836)

The new manuscript needs to be proofread for English, as the English is good but sometimes a bit confusing. 

Ans: As per the reviewer's suggestion, we have incorporated necessary changes to make the manuscript more comprehensible.

I absolutely applaud the authors for their enormous effort to make their manuscript significantly better. They did not try to do just the bare minimum - instead they did an excellent job of following the suggestions of the reviewers. Well done~!

Ans: We thank the reviewer for the encouragement and the applaud. It surely humbles us that our efforts have been acknowledged. We again thank the reviewer for their guidance.